# Targeting cardiomyocyte ADAM10 ectodomain shedding promotes survival early after myocardial infarction

Erik Klapproth [1], Anke Witt[2], Pauline Klose[1], Johanna Wiedemann[1], Nikitha Vavilthota[1], Stephan R. Künzel[1], Susanne Kämmerer[1], Mario Günscht [1], David Sprott[2], Mathias Lesche [3], Fabian Rost [3], Andreas Dahl [3], Erik Rauch[4], Lars Kattner[4], Silvio Weber[1], Peter Mirtschink[5], Irakli Kopaliani[2], Kaomei Guan [1], Kristina Lorenz [6,7], Paul Saftig [8], Michael Wagner [1,9] & Ali El-Armouche [1] ✉

After myocardial infarction the innate immune response is pivotal in clearing of tissue debris as well as scar formation, but exaggerated cytokine and chemokine secretion with subsequent leukocyte infiltration also leads to further tissue damage. Here, we address the value of targeting a previously unknown a disintegrin and metalloprotease 10 (ADAM10)/CX3CL1 axis in the regulation of neutrophil recruitment early after MI. We show that myocardial ADAM10 is distinctly upregulated in myocardial biopsies from patients with ischemia-driven cardiomyopathy. Intriguingly, upon MI in mice, pharmacological ADAM10 inhibition as well as genetic cardiomycyte-specific ADAM10 deletion improves survival with markedly enhanced heart function and reduced scar size. Mechanistically, abolished ADAM10-mediated CX3CL1 ectodomain shedding leads to diminished IL-1β-dependent inflammation, reduced neutrophil bone marrow egress as well as myocardial tissue infiltration. Thus, our data shows a conceptual insight into how acute MI induces chemotactic signaling via ectodomain shedding in cardiomyocytes.

Myocardial infarction (MI) is a leading cause of cardiac death and heart failure worldwide[1]. Inflammation is essential for clearing cellular debris and scarring after MI, but it also contributes to exaggerated myocardial tissue damage[2,3]. The degree of inflammation following MI largely depends on the number of infiltrating leukocytes[4,5]. Hence, reducing leukocyte infiltration to a level that ensures adequate infarct area healing but prevents detrimental excessive immune response is an intriguing future option for patients after MI[6].

Preclinical data indicate that pharmacological clearance of circulating cytokines after MI is a promising strategy[4,7]. However, clinical trials like CANTOS, ASSAIL-MI, RESCUE, COLCOT or CIRT on the one hand showed generally favorable outcomes but on the other hand also severe side effects such as fatal infection and sepsis[8–12]. This inspired us with the idea to move away from sole targeting of pleiotropic cytokines and chemokines and instead to intervene further upstream at potentially more tolerable targets to enable a

[1]Institute of Pharmacology and Toxicology, Faculty of Medicine Carl Gustav Carus, Technische Universität Dresden, Dresden, Germany. [2]Department of Physiology, Faculty of Medicine Carl Gustav Carus, Technische Universität Dresden, Dresden, Germany. [3]DRESDEN-concept Genome Center, Center for Molecular and Cellular Bioengineering, Technische Universität Dresden, Dresden, Germany. [4]Endotherm GmbH, Saarbruecken, Germany. [5]Institute of Clinical Chemistry and Laboratory Medicine, Faculty of Medicine Carl Gustav Carus, Technische Universität Dresden, Dresden, Germany. [6]Institute of Pharmacology and Toxicology, Julius-Maximilians-University of Würzburg, Würzburg, Germany. [7]Leibniz-Institut für Analytische Wissenschaften -ISAS- e.V., Dortmund, Germany. [8]Biochemical Institute, Christian-Albrechts-Universität Kiel, Kiel, Germany. [9]Rhythmology, Clinic of Internal Medicine and Cardiology, Heart Center Dresden, Technische Universität Dresden, Dresden, Germany. ✉e-mail: ali.el-armouche@tu-dresden.de

pronounced anti-inflammatory intervention for effective and tolerable cardioprotection.

Metalloproteases of the ADAM (a disintegrin and metalloprotease) family are membrane-anchored glycoproteins with diverse functions, including critical roles in fertilization, neurogenesis, and angiogenesis[13–16]. Ectodomain shedding of membrane-bound proteins is mainly mediated by ADAM10 and ADAM17. It is a post-translational regulatory process that coordinates leukocyte recruitment, attachment and transendothelial migration via proteolytic cleavage of cell surface bound cytokines (TNFα), chemokines (CXCL16 and CX3CL1) and their corresponding receptors (TNFRI/II, IL-1R2, IL-6R)[17–20].

Here we demonstrate ADAM10-mediated ectodomain shedding of CX3CL1 (also known as fractalkine) to critically modulate the transition between the pro-inflammatory and pro-reparatory phases via neutrophil but intriguingly not macrophage recruitment after MI. We show a small molecule inhibitor of ADAM10 to therapeutically target the ADAM10/CX3CL1 axis for improved survival and preserved cardiac function after MI in mice. This identifies a hitherto unknown and druggable axis, which may close a clinically highly relevant therapeutic gap early after MI with appropriate efficacy and safety.

## Results

### ADAM10 expression is elevated following experimental MI as well as in patients with ischemic cardiomyopathy

The presence of neutrophils and macrophages is not only necessary for proper scar formation after myocardial injury but also strongly correlates with excess secretion of pro-inflammatory cytokines, cardiac damage, and poor prognosis[21]. Hence, targeting an exaggerated neutrophil response is considered a promising strategy to reduce post-MI cardiac damage and prevent the progress to HF[22].

We hypothesized that ectodomain shedding via ADAMs is associated with leukocyte recruitment to injured cardiac tissue. Consequently, we subjected 8-week-old C57BL/6J mice to sham operation or ligation of the left anterior descending artery (LAD) for induction of MI. Three days after LAD ligation, the infarcted region was excised together with the infarct border zone, analyzed via western blot, and compared to similarly processed sham samples. ADAM10 was significantly upregulated after MI (2.4-fold), while ADAM8, ADAM12, ADAM17 and ADAM19 were unchanged (Fig. 1a, b). To validate the translational relevance of the uncovered ADAM10 upregulation, we analyzed ADAM10 expression in tissue lysates from patients with ischemic (ICM) and dilated cardiomyopathy (DCM) or non-failing hearts (NF). Western blot analysis revealed a significantly higher expression of ADAM10 in ICM (2.5-fold) but not DCM patients (Fig. 1c and Supplementary Fig. 1a). Moreover, mRNA expression analysis displayed a high correlation of *ADAM10* expression with the expression of the HF markers *NPPA*, encoding for atrial natriuretic peptide (ANP), and *NPPB* encoding for brain natriuretic peptide (BNP), in ICM patients (Fig. 1d, e). The elevated ADAM10 levels were preserved in mice 14 days after MI (Fig. 1f, g). Together, these data argue for specific upregulation of ADAM10 in the context of chronic ischemia as well as MI and suggest a possible connection between ADAM10 expression and HF severity in ICM.

We next determined the cardiac cell type responsible for enhanced ADAM10 expression upon ischemia. Histological analysis of sham-operated and LAD-ligated mice 14 days after infarction and screening of a panel of human induced pluripotent stem cell-derived cardiomyocytes (iCM), mouse immortalized cardiomyocytes (HL-1), human atrial and ventricular fibroblasts (HAF and HVF) and human endothelial cells (EC) revealed that cardiomyocytes as well as endothelial cells exhibit the highest ADAM10 expression (Fig. 1h and Supplementary Fig. 1b–g). This observation was confirmed by analysis of publicly available human single-cell sequencing gene expression data of the "Heart Cell Atlas"[23] (Fig. 1i and Supplementary Fig. 1d). However, cardiomyocytes were the only cell type where ADAM10 expression was

induced upon infarction (Fig. 1h). Mimicking hypoxia with the HIF hydroxylase inhibitor dimethyloxaloylglycine (DMOG) led to induction of mature ADAM10 in all tested cardiomyocyte-samples, which was less pronounced in endothelial cells and absent in cardiac fibroblasts (Supplementary Fig. 1h, i). Together, our data show a marked upregulation of cardiomyocyte ADAM10 expression in response to ischemia.

### Therapeutic potential of A10i in infarcted hearts

Since ADAM10 protein expression is upregulated 3 days after experimental MI, we tested the therapeutic potential of ADAM10 inhibition using the hydroxamate ADAM10 inhibitor GI254023X (A10i)[24]. Following LAD ligation, 8-week-old C57BL/6J mice were treated for up to 14 days with 100 mg/kg A10i or DMSO control using osmotic minipumps. Pumps were not primed before implantation, which leads to a delayed onset of substance release (4–6 h) mimicking the potential clinical scenario of treatment after onset of MI. After 3, 14 and 28 days, echocardiography was performed before the animals were sacrificed to collect serum and heart tissue (Fig. 2a). Kaplan–Meier analysis showed a significantly improved 14 day-survival of A10i treated mice compared to DMSO controls (Fig. 2b). Echocardiography 3 days post MI showed impaired cardiac function in control mice as evidenced by reduced fractional area shortening (FAS) as well as ejection fraction (EF) and dilation of the left ventricle (LVID), which progressed until day 28 (Fig. 2c–e). On the contrary, A10i treated mice had a significantly improved EF already 3 days after MI while FAS and LVID were not yet significantly improved when compared to controls (Fig. 2c). Strikingly, 14 and 28 days after infarction, echocardiographic analysis demonstrated a pronounced cardioprotective effect of A10i showing significantly improved FAS, EF and LVID in A10i treated mice compared to controls (Fig. 2d–f and Supplementary Fig. 2a, b). Importantly, 28 days after infarction (which is 14 days after treatment discontinuation), mice that had received A10i had EF and LVID values that were not significantly different from sham-operated animals (Fig. 2e). Notably, improved cardiac function was accompanied by a significantly reduced scar size in A10i treated hearts as revealed by histological analyses 28 days after ischemic injury (Fig. 2g, h). Cardiomyocyte cross sectional area was significantly reduced by A10i 28 days after MI compared to controls (Supplementary Fig. 2c). Taken together, these findings support the view that ADAM10 critically drives post-MI scar extent, LV remodeling and cardiac function.

### A10i treatment reduces post-MI neutrophil recruitment

We next analyzed the transcriptome of the infarct and border zone of 8-week-old, LAD ligated, DMSO and A10i treated C57BL/6J mice. Three days after MI, both treatment groups separated well in the principal component analysis and were clearly distinct from sham operated mice (Fig. 3a). Volcano plot analysis revealed 200 significantly upregulated and 206 significantly downregulated genes in the samples treated with A10i as compared to DMSO controls using a false discovery rate of 0.01 (Fig. 3b). Strikingly, functional annotation clustering and gene set enrichment analysis revealed that genes with significantly decreased expression upon A10i treatment were involved in biological processes like inflammatory response, neutrophil migration and chemotaxis (Fig. 3c–f). Annotation clustering identified chemokine and cytokine receptor binding and activity as well as metalloendopeptidase activity as significantly downregulated upon A10i treatment. These data indicate A10i to suppress post-infarction cardiac inflammation by ameliorating chemokine/cytokine activity. In depth analysis of the regulated gene clusters showed significant downregulation of the chemokines *Cxcl1*, *Cxcl3* and *Cxcl5* as well as the neutrophil marker genes *S100a8* and *S100a9* and the proinflammatory genes *interleukin (Il)1b* and *Il11* in heart tissue samples of A10i treated mice as compared to DMSO controls 3 days after MI (Fig. 3g), which was confirmed by quantitative real-time PCR (Supplementary Fig. 3a).

Moreover, several genes coding for ADAMs and MMPs like *Adam19*, *Adamts9* and the neutrophil associated *Mmp9* were significantly lower expressed when mice were treated with A10i (Fig. 3g). Next, we validated the mechanistic relevance of the downregulated chemokines by determining the respective serum levels 3 days after MI. We found an abrogation of serum IL-1β levels as well as a striking 50% reduction in serum CX3CL1, while the levels of IL-6, CXCL1, CXCL5 and CXCL16 were not significantly altered (Fig. 3h). We next assessed cardiac expression of CX3CL1 and IL-1β at 3, 14 and 28 days post myocardial infarction using immunoblot analysis. Evaluating full-length (non-shed) CX3CL1 using an N-terminal antibody, we found CX3CL1 to be decreased at 3 days post MI, consistent with increased shedding of ADAM10, that was no longer detectable on day 14 and day 28 (Fig.3i, j). In contrast, IL-1β remained elevated through day 28 (Fig. 3i, k). Both changes were sensitive to treatment with A10i (Fig. 3i–k). Other substrates of ADAM10 and also ADAM17 were not affected by A10i

(Supplementary Fig. 3b). CX3CL1 shedding was not only dysregulated by experimental heart injury but also in patients with ICM (Fig. 3l, m). Together, these results point at an ADAM10-driven mechanism that governs post-infarction inflammation potentially via downregulation of neutrophil recruitment to the injured heart by regulation of chemokine signaling.

To test this hypothesis, we used flow cytometry to evaluate post-infarction leukocyte counts in blood and heart tissue samples (Fig. 4a–c). Three days after MI, blood sample analysis showed a striking reduction of the neutrophil population (Fig. 4b) with A10i treatment. While eosinophil and basophil populations showed a tendency to reduced abundance, monocyte and lymphocyte counts remained unaffected by A10i (Fig. 4b). In line, we found significantly decreased neutrophil counts (CD45$^+$CD11b$^+$LY6G$^+$) in the infarct and border zone tissue of A10i treated mice while populations of eosinophils (CD45$^+$CD11b$^+$LY6G$^-$SigF$^+$), monocytes

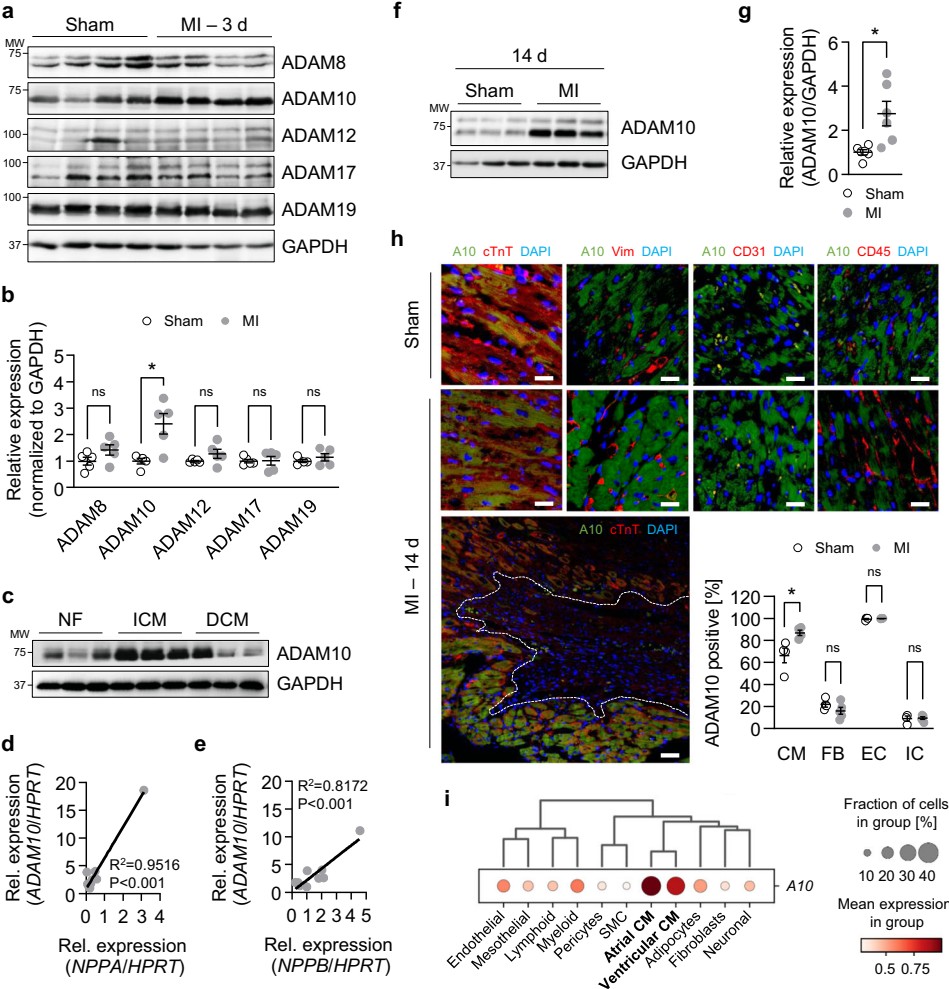

**Fig. 1 | ADAM10 is upregulated in experimental heart injury as well as patients with ischemic cardiomyopathy and correlates with ANP/ BNP expression.**
**a** Western blot analysis and **b** quantification of indicated ADAMs in heart tissue lysates of sham-operated (Sham) and LAD-ligated (MI) mice 3 days after infarction (*n* = 5, mean ± SEM, ns not significant, *\*P* < 0.05, two-tailed Mann–Whitney test). Exact *P*-values: ADAM8, 0.150794. ADAM10, 0.015873. ADAM12, 0.150794. ADAM17, 0.690476. ADAM19, 0.690476. **c** Western blot analysis of ADAM10 in heart tissue lysates from patients with ischemic cardiomyopathy (ICM) and dilated cardio-myopathy (DCM) versus non-failing controls (NF) (*n* = 6). Pearson's correlation of ADAM10 mRNA expression against expression of **d** NPPA (ANP) and **e** NPPB (BNP) (*n* = 12). Exact *P*-values: NPPA, <0.0001. NPPB, 0.0008. **f** Western blot analysis and **g** quantification of ADAM10 in heart tissue lysates of mice 14 days after sham-

operation (Sham) and LAD-ligation (MI) (*n* = 6, mean ± SEM, *\*P* = 0.011, two-tailed *t* test). **h** Immunofluorescence images and quantification of ADAM10 (A10), cardiac Troponin T (cTnT, cardiomyocytes, CM), Vimentin (Vim, fibroblasts, FB), CD31 (endothelial cells, EC) and CD45 (immune cells, IC) stained heart tissue sections of sham-operated (Sham) and LAD-ligated (MI) mice 14 days after surgery. Nuclei are stained with DAPI. Representative images are shown. Scale bar – upper panel, 20 μm. Scale bar – lower panel, 10 μm. Dotted line indicates the infarct area. *n* = 4 per group, mean ± SEM, ns not significant, *\*P* < 0.05, two-tailed Mann–Whitney test. Exact *P*-values: CM, 0.028571. FB, 0.190476. EC, 0.714286. IC, 0.904762. **i** Gene expression signature of ADAM10 (A10) in indicated cell types of human hearts. Publicly available single cell sequencing data of the "Heart Cell Atlas" were analyzed. Source data are provided as a Source Data file.

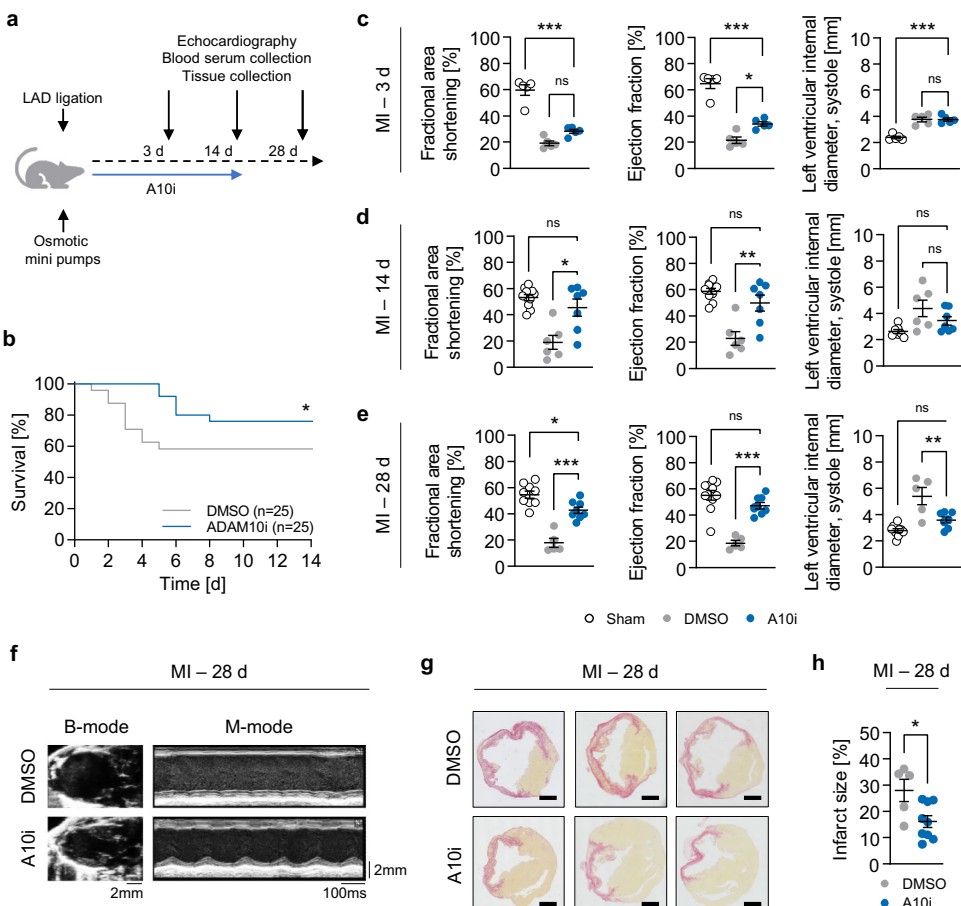

**Fig. 2 | Pharmacological ADAM10 inhibition improves survival and augments cardiac function after experimental infarction. a** Treatment scheme. Following LAD-ligation (MI) or sham-operation (Sham, $n = 25$), mice were treated with GI254023X (A10i, 100 mg/kg, $n = 30$) or DMSO ($n = 30$) for up to 14 days using osmotic minipumps. Final echocardiography and sample collection was performed 3, 14 and 28 days after LAD ligation. **b** Kaplan–Meier survival curves of LAD-ligated, GI254023X (A10i) and DMSO treated mice. Mice enrolled in the 3-day group were not included in the analysis ($n = 25$ per group, *$P = 0.0463$, Log-rank test). Echocardiographic assessment of fractional area shortening, ejection fraction and left ventricular end-systolic interior diameter **c** 3 days (Sham, $n = 5$. DMSO, $n = 5$. A10i,

$n = 5$), **d** 14 days (Sham, $n = 10$. DMSO, $n = 6$. A10i, $n = 7$) and **e** 28 days (Sham, $n = 9$. DMSO, $n = 5$. A10i, $n = 8$) after myocardial infarction (mean ± SEM, *$P < 0.05$, **$P < 0.01$, ***$P < 0.001$, one-way ANOVA with Tukey's posttest, exact $P$-values are provided in the Source Data file). **f** Representative end-systolic B-mode and M-mode echocardiograms of GI254023X (A10i) and DMSO treated mice 28 days after myocardial infarction. **g** Picro sirius red staining of heart tissue sections of GI254023X (A10i) and DMSO treated mice 28 days after infarction and **h** quantification of the size of the fibrotic scar (DMSO, $n = 5$. A10i, $n = 8$, mean ± SEM, *$P = 0.0246$, two-tailed $t$ test). Source data are provided as a Source Data file.

(CD45$^+$CD11b$^+$LY6G$^-$SigF$^-$F4/80$^-$LY6C$^+$) and macrophages (CD45$^+$CD11b$^+$LY6G$^-$SigF$^-$F4/80$^+$) were not significantly different from DMSO treated samples (Fig. 4a, c). Importantly, in sham operated animals blood leukocyte counts as well as leukocyte tissue infiltration were not affected by A10i treatment (Supplementary Fig. 4a, b). In summary, these data reveal that ADAM10 inhibition following MI reduces neutrophil-driven inflammation for improved survival and cardioprotection.

### Treatment with A10i for 3 days after MI is as effective as treatment for 14 days

The fact that the effects of MI on CX3CL1, neutrophil infiltration, IL-1β and pump function were visible as soon as day three post MI led us to the hypothesis that treatment with A10i for 3 days after MI may be sufficient for the therapeutic effect. Consequently, we compared cardiac function in mice treated for 3 or 14 days at day 14 and 28 after MI (Fig. 4d). At both time points, cardiac function in animals treated for 3 days was significantly improved and virtually indistinguishable from the animals treated for 14 days (Fig. 4e, f). This opens the opportunity for short-term A10i treatment early after MI to limit infarction size.

### An ADAM10/CX3CL1 signaling axis in cardiomyocytes regulates neutrophil migration

Our in vivo data indicated a ADAM10/CX3CL1 axis to control neutrophil infiltration upon myocardial infarction. Since cardiac ADAM10 is mostly expressed in cardiomyocytes (Fig. 1h, i and Supplementary Fig. 1c, e, f) and CX3CL1 was the most significantly reduced chemokine in serum samples of A10i treated mice (Fig. 3h), we next examined whether ADAM10 regulates neutrophil migration directly via ectodomain shedding of CX3CL1 in cardiomyocytes. Therefore, we first determined CX3CL1 expression in different cardiac cell types. Interestingly, histological analysis in murine heart tissue as well as western blot and qPCR analysis in human iCMs, atrial and ventricular fibroblasts (HAF, HVF) and endothelial cells (EC) revealed CX3CL1 to be most highly expressed in cardiomyocytes and endothelial cells (Fig. 5a, b and Supplementary Fig. 5a–d). Ectodomain shedding by ADAM10 for regulation of substrate activity, such as described for CX3CL1[25], requires at least transient physical interaction between the protease and its substrate[17]. Hence, we performed co-immunoprecipitation experiments in HL-1 cells and found endogenous CX3CL1 to co-precipitate with endogenous ADAM10 (Fig. 5c). Moreover, CX3CL1 ectodomain shedding to the cell culture supernatant was significantly

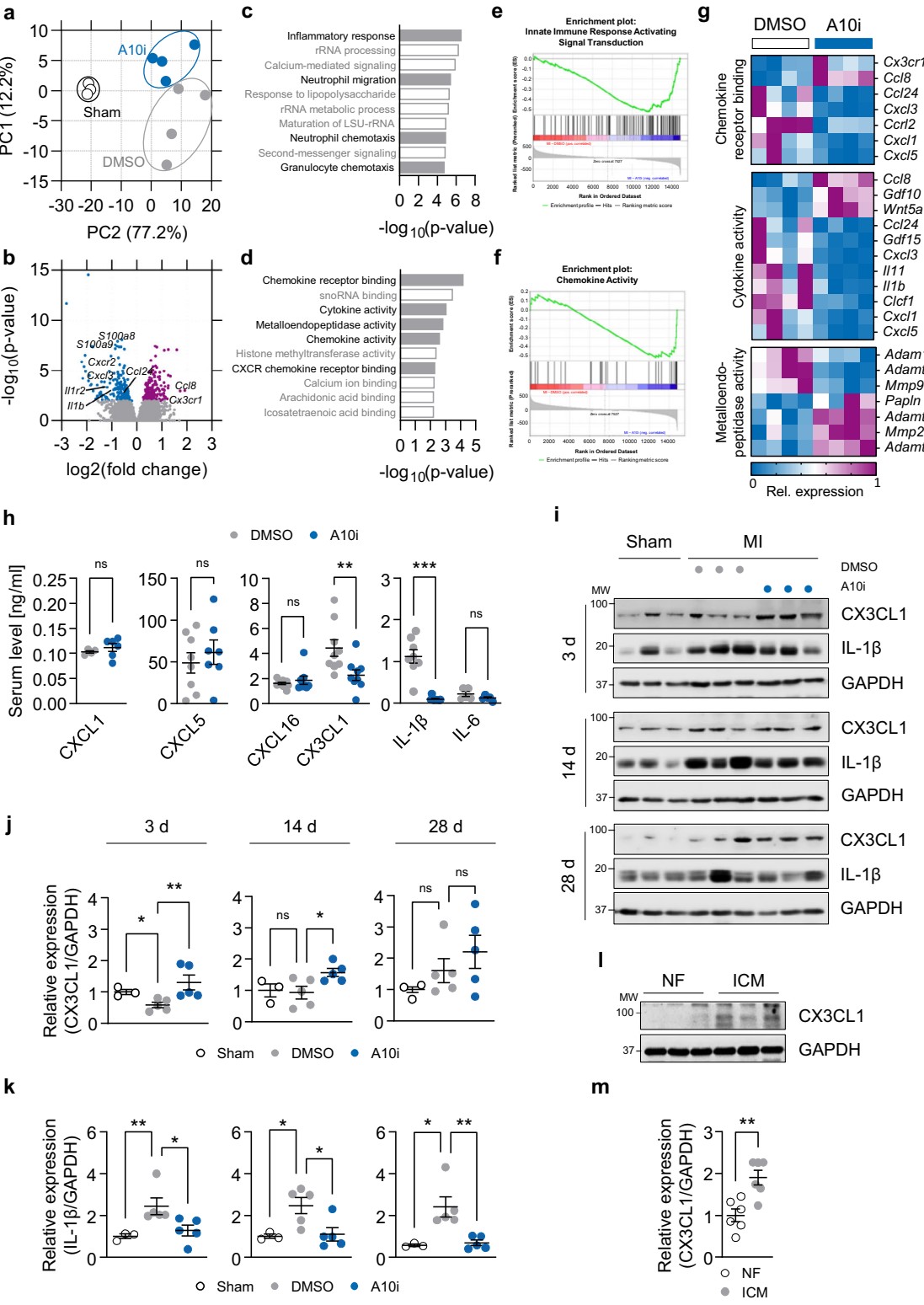

increased when HL-1 mouse cardiomyocytes were exposed to 3 h of hypoxia (Fig. 5d–f). Parallel treatment of HL-1 cells with A10i (10 μM), in turn, maintained CX3CL1 expression at basal levels (Fig. 5d–f).

To prove that the demonstrated cardiomyocyte-specific ADAM10/CX3CL1 axis is capable of modulating leukocyte migration, we employed transwell migration assays. To generate chemoattracting media for leukocyte migration analysis, we treated HL-1 cells with either DMSO or A10i 24 h prior to normoxic and hypoxic conditioning. After 3 h of normoxia/hypoxia, supernatant of the

treated HL-1 cells was harvested and used as chemoattractant in transwell assays. Thirty thousand freshly isolated mouse bone marrow derived neutrophils or macrophages were seeded in RPMI medium without additives in the upper chamber of the transwell plates and incubated for 12 h under standard culture conditions (Fig. 5g). Following incubation, cells were fixed, DAPI stained, and cell counts per field were analyzed by fluorescence microscopy. We found that 3 h of hypoxia significantly stimulated neutrophil migration, which was suppressed by addition of A10i (Supplementary

**Fig. 3 | ADAM10 inhibition after MI reduces neutrophil chemotaxis-associated gene expression and CX3CL1 serum levels. a** Principal components analysis (PCA) of transcriptomes of sham-operated (Sham, $n = 3$), GI254023X (A10i, $n = 4$) and DMSO ($n = 4$) treated mice 3 days after myocardial infarction. Principal components were computed for each sample using gene expression. The first two principal components (PC1 and PC2) are shown. The percent of variance explained by each component is reported in parentheses. **b** Volcano plot representing the expression changes of all genes. Significantly down- and upregulated genes (FDR < 0.01) are colored navy and magenta, respectively. Genes that do not show significant expression changes are colored gray. Examples of neutrophil chemotaxis-associated genes are labeled with gene names. *P*-values were calculated using the R packages DESeq2 R (v1.30.1)[52] and IHW (1.18.0)[53]. Functional annotation clustering for the gene ontology (GO) terms **c** biological process and **d** molecular function of genes with significantly decreased expression upon A10i treatment (Fisher exact test with Benjamini-Hochberg posttest). **e, f** Gene set enrichment analysis of genes with significantly decreased expression upon A10i treatment

performed on GO terms. Gene set enrichment analysis was performed using the GSEA tool (v4.1.0) of the Broad Institute[55] and the DESeq2 R package (v1.30.1)[52]. **g** In-depth analysis of the significantly downregulated genes of the indicated functional annotation clusters. **h** Quantification of CXCL1 ($n = 4$ vs. 5, $P = 0.380952$), CXCL5 ($n = 8$ vs. 8, $P = 0.633877$), CXCL16 ($n = 10$ vs. 8, $P = 0.572604$), CX3CL1 ($n = 9$ vs. 9, $P = 0.00399$), IL-1β ($n = 8$ vs. 8, $P = 0.000155$) and IL-6 ($n = 6$ vs. 5, $P = 0.452381$), serum levels in A10i and DMSO treated mice 3 days after MI (mean ± SEM, ns not significant, **$P < 0.01$, ***$P < 0.001$, two-tailed Mann–Whitney test). **i** Western blot analysis and **j, k** quantification of **j** CX3CL1 and **k** IL-1β in heart tissue lysates of sham-operated (Sham) and LAD-ligated (MI) mice 3 days after infarction ($n = 5$, mean ± SEM, ns not significant, *$P < 0.05$, **$P < 0.01$, Kruskal–Wallis test with Dunn's posttest, exact *P*-values are provided in the Source Data file). **l** Western blot analysis and **m** quantification of CX3CL1 in heart tissue lysates from patients with ischemic cardiomyopathy (ICM) and non-failing controls (NF) ($n = 6$, mean ± SEM, **$P = 0.0031$, two-tailed *t* test). Source data are provided as a Source Data file.

Fig. 5e). This indicated that ADAM10 inhibition in cardiomyocytes regulates leukocyte migration by potentially altering the release of chemoattractants during hypoxic conditioning. To evaluate whether this ADAM10-controled mechanism was due to reduced ectodomain shedding of CX3CL1 upon A10i treatment, we applied the same experimental setting with HL-1 cells that were depleted for CX3CL1 using small interfering RNA (siRNA) (Fig. 5h–k). Consistently, A10i treatment resulted in reduced neutrophil migration as compared to DMSO treated cells. Small interfering RNA-mediated depletion of CX3CL1 led to a decreased neutrophile migration capacity that was comparable to A10i treatment (Fig. 5j). Most importantly, when we combined CX3CL1 knockdown with A10i, we could not detect any additional effect on neutrophil migration (Fig. 5j). Moreover, sole or combined A10i treatment and CX3CL1 depletion in HL-1 cardiomyocytes did not affect the migration capacity of mouse bone marrow derived macrophages (Fig. 5k) arguing for a jointly used signaling axis of cardiomyocyte ADAM10 and CX3CL1 to specifically modulate neutrophil chemotaxis.

**Cardiomyocyte-specific ADAM10 knockout protects from ischemic injury similar to A10i**

Deficiency of ADAM10 results in prenatal lethality around embryonic day 9.5 due to severe growth retardation and defects in the developing central nervous and cardiovascular systems[19]. To prove that the effects of A10i were in fact due to reduced ADAM10 activity in cardiomyocytes, we generated a mouse model with cardiomyocyte-specific deletion of ADAM10 via αMHC-mediated Cre-lox gene recombination making use of floxed ADAM10 mice[13] (Supplementary Fig. 6a). Histopathological characterization in 6-month-old mice revealed no overt signs of cardiac hypertrophy or fibrosis when cardiomyocyte-specific ADAM10 knockout (KO) mice were compared to ADAM10fl/fl (WT) littermates (Supplementary Fig. 6b, c). In echocardiography, 6-month-old ADAM10 KO mice did not show differences in EF, FAS or LVID compared to WT littermates (Fig. 6a). We next investigated the effects of the ADAM10 KO in MI. As soon as 3 days after LAD ligation, ADAM10 KO mice showed a significantly higher EF compared to WT controls. By day 14, FAS and EF as well as LVID were improved as compared to WT controls (Fig. 6b–d) validating the demonstrated cardioprotective effects of A10i. Importantly, and similar to the results of A10i, ADAM10 KO resulted in significantly reduced shedding of CX3CL1 in the infarct and border zone and reduced cardiac expression of IL-1β 3 days after infarction (Fig. 6e, f). ADAM10 expression was significantly reduced. Also, cardiac neutrophil invasion was significantly lower in ADAM10 KO 3 days after MI as were blood neutrophil counts (Fig. 6g, h and Supplementary Fig. 7). Together our data show that cardiomyocyte-specific genetic ablation of ADAM10 results in reduced CX3CL1 chemotaxis and repression of exaggerated neutrophil infiltration after MI.

## Discussion

We demonstrate that inhibition of the sheddase ADAM10 early after MI by either a pharmacological inhibitor or genetic ablation improves survival and cardiac function and alleviates progression to severe heart failure. Moreover, our study establishes a cardiomyocyte sheddase-driven mechanism that specifically controls neutrophil egress from bone marrow and infiltration of cardiac tissue through CX3CL1/fractalkine release by the heart.

We found that myocardial ADAM10 is markedly upregulated in biopsies from patients with ischemia-driven cardiomyopathy. Given the fact that ADAM10 is a critical factor in embryonic development[19,20,26,27], it is conceivable that its upregulation is part of the phenomenon of the activation of the fetal gene program occurring after significant cardiac injury. This return to the fetal gene program is considered as detrimental and is ubiquitously used as a biomarker of cardiac hypertrophy and as a hallmark for progressive decline in cardiac function[28]. Moreover, *ADAM10* expression correlates well with two commonly measured fetal genes and markers of heart failure, *NPPA* and *NPPB*[29].

We found that targeting the ADAM10/CX3CL1 axis modulates the transition between the pro-inflammatory and pro-reparatory phases via neutrophil but intriguingly not macrophage or eosinophil recruitment. This is consistent with the fact that neutrophils are the first dominant leukocyte subset that is recruited to the site of MI. Their numbers peak in the first 3 days after cardiac injury. Only thereafter, monocytes and monocyte-derived macrophages become the predominant infiltrating cell types[30]. Nevertheless, monocytes and macrophages have in the past been considered the crucial leukocyte subset after myocardial infarction, and much attention has been focused on deciphering the mechanisms that drive monocyte recruitment for cardioprotection[6,30]. Thus, our results indicate a unique and specific role for cardiomyocyte ADAM10/CX3CL1 in neutrophil infiltration and underscore the dangers of escalated neutrophil recruitment for cardiac function early after MI.

Importantly, our data show that short-term A10i treatment after MI maintains low levels of IL-1β even after discontinuation, providing the opportunity for therapeutic lowering of this proinflammatory cytokine via transient suppression of ectodomain shedding of chemokines. We validated this hypothesis by demonstrating that heart function upon treatment with A10i for 3 days after MI is identical to 14-day treatment. This is of particular interest, as IL-1β is not only crucially involved in transient activation and recruitment of immune cells but is also involved in the formation of an immunologic memory of the innate immune system[31]. This may link the ADAM10/CX3CL1 axis to the concept of trained innate immunity, in which innate immune cells acquire a long-lasting proinflammatory phenotype after initial inflammatory stress and then worsen chronic inflammatory conditions[32,33]. Thus, targeted inhibition of the ADAM10/CX3CL1 axis could interfere

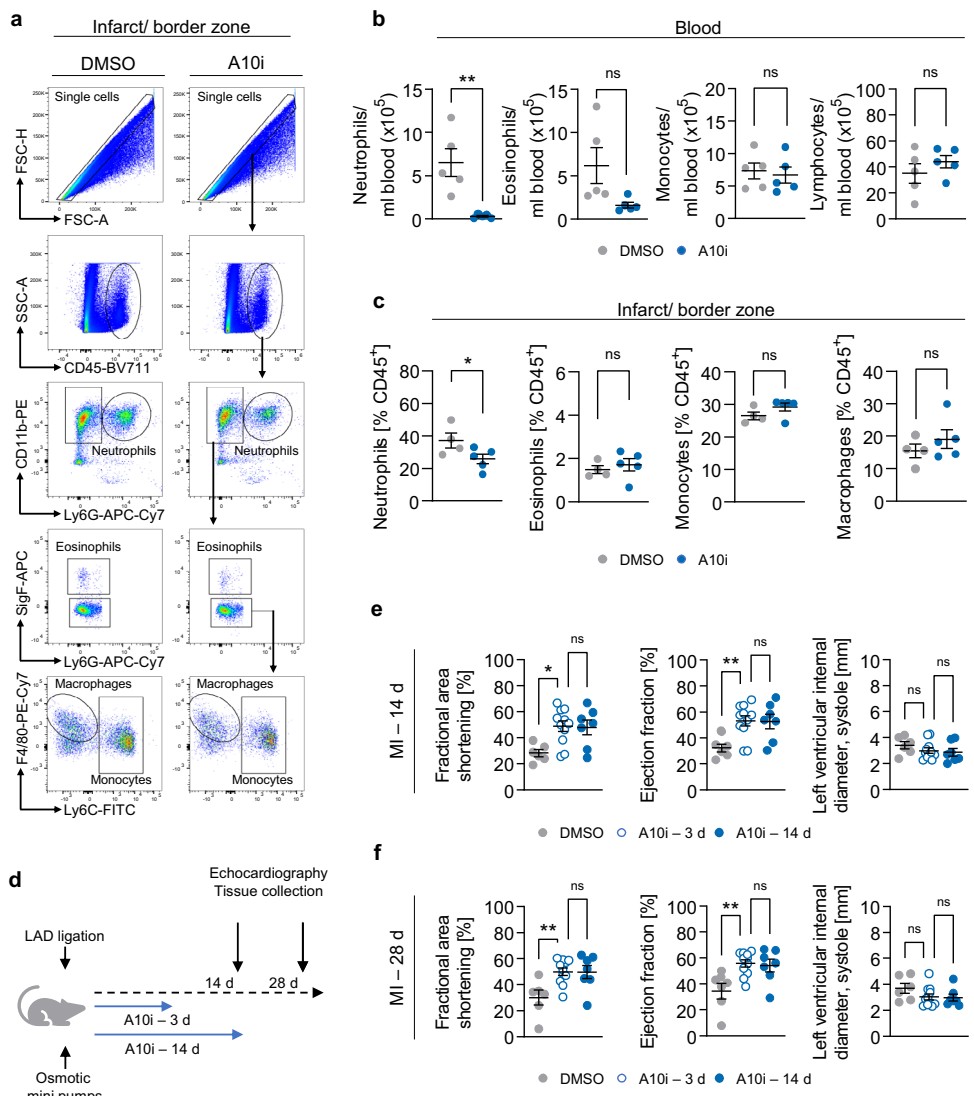

**Fig. 4 | ADAM10 inhibition reduces neutrophil bone marrow egress and heart tissue infiltration following myocardial infarction. a** Gating strategy for identification of neutrophils (CD45[+], CD11b[+], Ly6G[+]), eosinophils (CD45[+], CD11b[+], Ly6G[−], SigF[+]), monocytes (CD45[+], CD11b[+], Ly6G[−], SigF[−], F4/80[−], Ly6C[+]) and macrophages (CD45[+], CD11b[+], Ly6G[−], SigF[−], F4/80[+]) in the infarcted area/ infarct border zone of A10i (n = 5) and DMSO (n = 4) treated mice 3 days after infarction. **b** Analysis of leukocyte counts in blood samples of GI254023X (A10i, n = 5) and DMSO (n = 5) treated mice 3 days after infarction (mean ± SEM, ns not significant, **P < 0.01, two-tailed t test). Exact P-values: Neutrophils, 0.0047. Eosinophils, 0.0613. Monocytes, 0.7339. Lymphocytes, 0.3485. **c** Analysis of leukocyte counts in the infarcted area/ infarct border zone of GI254023X (A10i, n = 5) and DMSO (n = 5) treated mice 3 days after infarction (mean ± SEM, ns not significant, *P < 0.05, two-tailed t test). Exact P-values: Neutrophils, 0.049. Eosinophils, 0.5605. Monocytes, 0.1624. Macrophages, 0.3721. **d** Treatment scheme. Following LAD-ligation (MI), mice were treated with GI254023X (A10i, 100 mg/kg) for 3 days (n = 12), 14 days (n = 7) or DMSO (n = 10) using osmotic minipumps. Final echocardiography and sample collection was performed 14 and 28 days after LAD ligation. Echocardiographic assessment of fractional area shortening, ejection fraction and left ventricular end-systolic interior diameter **e** 14 and **f** 28 days (DMSO, n = 6. A10i − 3 d, n = 11, A10i − 14 d, n = 7) after myocardial infarction (mean ± SEM, *P < 0.05, **P < 0.01, one-way ANOVA with Tukey's posttest). Source data and exact P-values are provided in the Source Data file.

with the dynamic events triggering detrimental trained immunity and chronic inflammation in ischemic heart disease.

In addition to CX3CL1 also other processes may contribute to the effects of A10i and ADAM10 KO. We therefore probed for a number of well-known ADAM10 (or ADAM17) targets which were not affected by pharmacological ADAM10 inhibition after MI. This does not exclude the involvement of other pathways not explored in our experiments. Moreover, single cell transcriptomics would allow a more comprehensive description of the actions of specific cell types in the processes occurring in the infarct and border zone.

Since inhibition of ADAM10-mediated ectodomain shedding of chemokines has led to markedly positive results after myocardial infarction, the question of transferability of these findings to other inflammation-driven diseases arises. In line with our data,

pharmacological or genetic ablation of ADAM10 in mouse models of traumatic brain injury and acute pulmonary inflammation revealed a critical role of ADAM10 for regulation of pro-inflammatory cytokine expression and chemotactic recruitment of neutrophils and monocytes. In both models, ADAM10-deficiency correlated significantly with tissue protection and improved organ function[34,35]. In rheumatoid arthritis, ADAM10-mediated shedding drives memory T-cell infiltration into rheumatoid joints promoting local inflammation and exacerbating joint injury[36]. Moreover, ADAM10/CX3CL1 serum levels are elevated in rheumatoid arthritis patients and positively predict treatment responsiveness to tocilizumab, a monoclonal antibody targeting the IL-6 receptor[37]. Monitoring ADAM10/CX3CL1 levels may therefore be of utility in stratifying patients who will benefit most from anti-inflammatory therapies in many pro-inflammatory diseases.

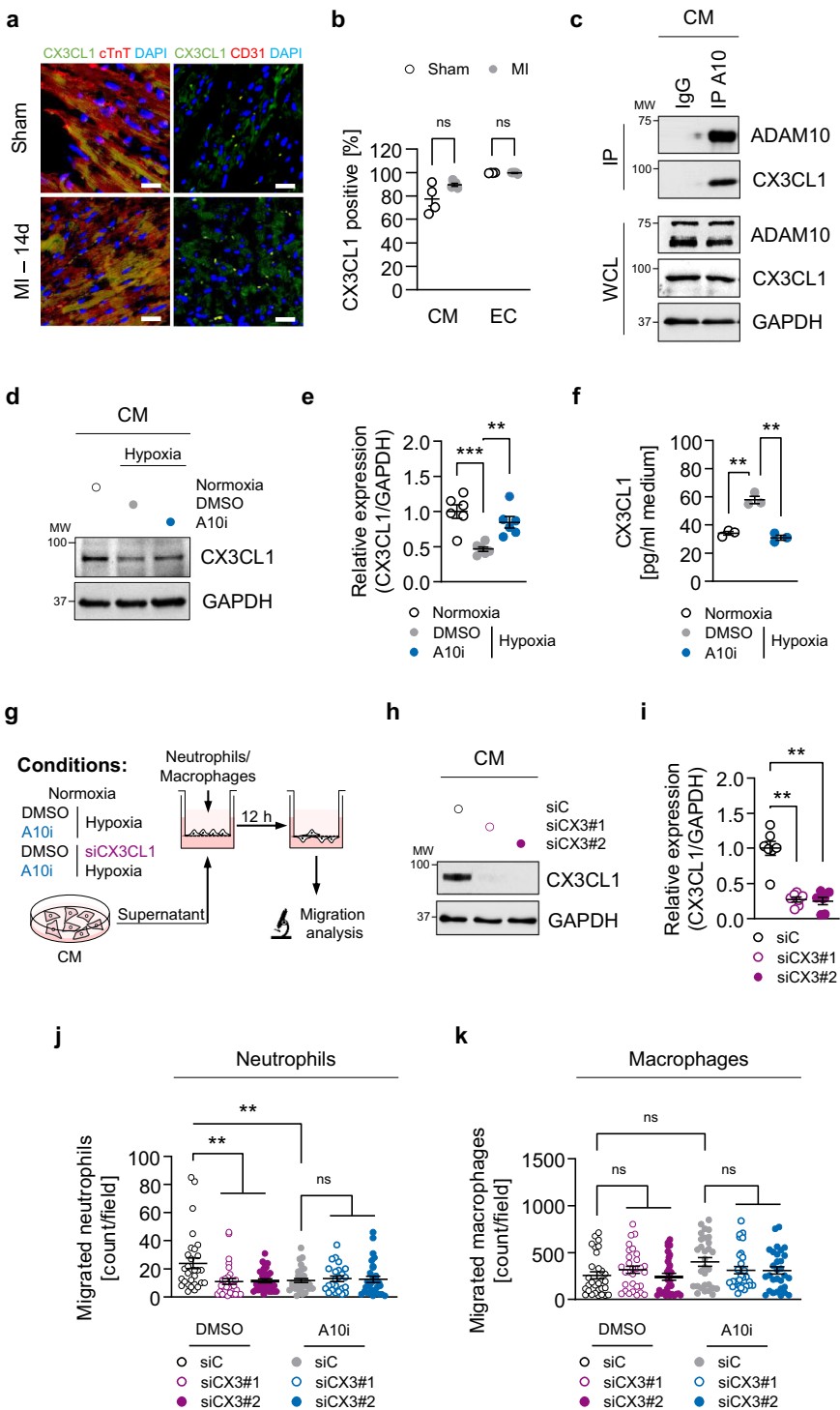

Our data are in line with clinical trials using inhibition of post-MI inflammation by the IL-6R blocking antibody tocilizumab as well as colchicine[8,12]. Of note, in the COLCOT trial, colchicine, which inhibits the migration of neutrophils and other leukocytes, led to favorable results especially when given early (within 3 days) after MI[38], which supports our hypothesis that anti-inflammatory intervention after MI as early as possible may be the most promising strategy. Our data indicate that treatment with A10i for only 3 days post MI may be sufficient for protection from excessive myocardial inflammation, leading to reduced infarct size as well as an expected favorable side effect profile with reduced risk of e.g. hepatotoxicity[39] and without prolonged immunosuppression and the risk of severe infection. The

increased ADAM10 expression in ICM patients makes it likely that ADAM10 is also involved in the pathophysiology of chronic cardiac ischemia. The importance of inflammation in this setting has been recently underlined in the CANTOS trial using IL-1β inhibition with canakinumab[10]. Thus, it seems worthwhile to test A10i also in this scenario in future studies.

Together, the ADAM10/CX3CL1 axis described here raises the intriguing possibility of therapeutic short-term inhibition of ADAM10 as a strategy to limit scar size and preserve cardiac function after acute MI. This may exert prolonged protective effects and at the same time limit side effects like immunosuppression to a short period of time. Moreover, the release of proinflammatory chemokines and cytokines

**Fig. 5 | Cardiomyocyte-specific ADAM10/CX3CL1 signaling regulates neutrophil migration. a** Immunofluorescence images and **b** quantification of CX3CL1, cardiac Troponin T (cTnT, cardiomyocytes, CM) and CD31 (endothelial cells, EC) stained heart tissue sections of sham-operated (Sham) and LAD-ligated (MI) mice 14 days after surgery. Nuclei are stained with DAPI. Representative images are shown. Scale bar, 20 μm. $n = 4$ per group, mean ± SEM, ns not significant, two-tailed Mann–Whitney test with Dunn's posttest. **c** Co-immunoprecipitation of endogenous ADAM10 (A10) and CX3CL1 from whole-cell lysates of HL-1 cells. Representative western blots are shown ($n = 4$). IgG, immunoglobulin G as a control. WCL, whole-cell lysate. **d** Western blot analysis of CX3CL1 and **e** quantification of CX3CL1 expression ($n = 6$ per group, mean ± SEM, ns not significant, **$P < 0.01$, ***$P < 0.001$, one-way ANOVA with Tukey's posttest) as well as **f** ectodomain shedding (CX3CL1 levels in supernatant) in normoxic as well as GI254023X (A10i) and DMSO treated hypoxic (1% $O_2$) mouse cardiomyocytes (HL-1) ($n = 3$ per group, mean ± SEM, ns not significant, **$P < 0.01$, Kruskal–Wallis test with Dunn's posttest, exact $P$-values are provided in the Source Data file). **g** Treatment scheme. HL-1 cells were cultured for 3 h under normoxic or hypoxic conditions in parallel to A10i or DMSO treatment in combination with unspecific control siRNA (siC) or CX3CL1 depletion (siCX3CL1). Supernatants were used in the lower chamber and $3 \times 10^5$ freshly isolated mouse bone marrow derived neutrophils or macrophages were seeded in the upper chamber of transwell plates (8 μm pore size) to determine migration capacity. The scheme was created using Servier Medical Art (available online: http://smart.servier.com). **h** Western blot analysis and **i** quantification of CX3CL1 expression in HL-1 cells treated for 48 h with unspecific control siRNA (siC) or CX3CL1-specific siRNA (siCX3#1 and siCX3#2). Representative western blots ($n = 3$ per group, mean ± SEM, **$P < 0.01$, Kruskal–Wallis test with Dunn's posttest) are shown. Exact $P$-values: siC vs. siCX3#1, 0.0043. siC vs. siCX3#2, 0.005. Transwell migration assays of bone marrow derived **j** neutrophils and **k** macrophages using the supernatants of HL-1 cells that were cultured under hypoxic conditions in parallel to A10i or DMSO treatment in combination with unspecific control siRNA (siC) or CX3CL1 depletion (siCX3#1 and CX3#2) as chemoattractant ($n = 3$ per group, mean ± SEM, ns not significant, **$P < 0.01$, Kruskal–Wallis test with Dunn's posttest). Source data and exact $P$-values are provided in the Source Data file.

by ADAM10 ectodomain shedding may be a mechanism for activation and recruitment of neutrophils also in other injury-related and inflammation-promoting diseases.

## Methods

### Human subjects
Collection of human tissue samples was approved by the ethics committee of the University Medical Center Goettingen (Az.:31/9/00, this approval is also applicable for the acquisition of samples at the University Hospital Regensburg). Heart samples were obtained from 12 patients with NYHA stage III-IV. Non-failing myocardium originated from 6 healthy donor hearts that could not be transplanted for technical reasons. All participants gave written informed consent. The study was conducted in accordance with the Declaration of Helsinki. Details on gender, age and clinical characteristics of these patients were published elsewhere[40].

### Animals
Animal facilities and experiments were authorized by the Landesdirektion Sachsen, Dresden, Germany, according to the German animal welfare regulations (No. TVV 54/2016, No. TVV16/2022) and comply with the ARRIVE guidelines and the guidelines from Directive 2010/63/EU of the European Parliament on the protection of animals used for scientific purposes. The mice were housed in rooms with a humidity of approximately 50%, a temperature of approximately 23 °C, and a 12-h light-dark rhythm. All mice were fed a standard diet and received water ad libitum. Mice were monitored at least once a week by specially trained animal care staff. The generation of the floxed ADAM10 mice was described previously[13,41]. To generate cardiomyocyte-specific ADAM10 KO mice, homozygous floxed mice (ADAM10$^{fl/fl}$) were crossed with α-myosin heavy chain (αMHC)-Cre transgenic mice to obtain mice heterozygous for the floxed ADAM10 allele and hemizygous for the αMHC-Cre allele, which were then crossed with ADAM10$^{fl/fl}$ mice to obtain ADAM10 KO mice. ADAM10$^{fl/fl}$ mice were used as wild-type control (WT). For inhibitor experiments, female wild-type C57BL/6J mice were obtained from Janvier Laboratories, Saint Berthevin Cedex, France.

### Cell cultures
HL-1 mouse cardiomyocytes were obtained from Sigma-Aldrich (SCC065). HL-1 cells were cultured in Claycomb medium (Sigma-Aldrich) supplemented with 10% fetal calf serum, 2 mM L-glutamine, 1% penicillin/streptomycin and 0.1 mM norepinephrine (all from Sigma-Aldrich) at 37 °C in a humidified atmosphere containing 5% $CO_2$. Human induced cardiomyocytes were generated from human induced pluripotent stem cells iWT.D2.1 (GOEi001-A.1; iCM1) and iBM76.3 (GOEi005-A.3; iCM2) as described[42] and maintained in RPMI 1640 with Glutamax, HEPES, and 2% B27 (all from ThermoFisher Scientific) for 60 days. Human ventricular fibroblasts (HVF) were purchased from Applied Biological Materials Inc. (T4038). Human atrial fibroblasts (HAF) and mouse ventricular fibroblasts were generated and characterized as published[43,44]. Fibroblasts were cultured in Dulbecco's modified Eagle's medium (ThermoFischer Scientific; plus 2 mM L-glutamine) supplemented with 10% fetal calf serum (Sigma-Aldrich) and 1% penicillin/streptomycin (ThermoFischer Scientific) at 37 °C in a humidified atmosphere containing 5% $CO_2$. Primary human umbilical vein endothelial cells (HUVEC) were kindly provided by C. Brunssen and isolated as described[45]. HUVEC were cultured in Endothelial Cell Growth Medium 2 (PromoCell) supplemented with 1% penicillin/streptomycin (ThermoFischer Scientific) at 37 °C in a humidified atmosphere containing 5% $CO_2$ and used for not more than two passages. Human THP-1 cells were purchased from Sigma-Aldrich (88081201). Mouse bone marrow derived neutrophils and macrophages were freshly isolated as detailed below. THP-1 and bone marrow derived cells were cultured in RPMI medium (ThermoFischer Scientific) supplemented with 10% fetal calf serum (Sigma-Aldrich) and 1% penicillin/streptomycin (ThermoFischer Scientific) at 37 °C in a humidified atmosphere containing 5% $CO_2$. All cell cultures were tested negative for mycoplasma contamination.

### Total protein extracts and Western Blotting
For protein lysates of heart tissue samples, ventricular myocardium was collected and homogenized using the TissueLyzer LT bead mill (Qiagen) for 4 min at a frequency of 50/s. Pulverized ventricular myocardium was lysed in a buffer containing 30 mmol/L Tris/HCl (pH 8.8), 5 mmol/L EDTA, 30 mmol/L NaF, 3% SDS, 10% glycerol (all from Sigma-Aldrich), cOmplete protease inhibitor cocktail including metalloprotease inhibitors (product No 04 693 124 001, Roche) and PhosSTOP phosphatase inhibitor cocktail (Roche). Cells were homogenized in RIPA buffer containing 150 mM NaCl, 50 mM Tris-HCl, 1 mM EDTA, 1% IGEPAL CA-630, 0.25% Na-deoxycholate, 0.1% SDS (all from Sigma-Aldrich), cOmplete protease inhibitor cocktail including metalloprotease inhibitors (product No 04 693 124 001, Roche) and PhosSTOP phosphatase inhibitor cocktail (Roche) by mechanical disruption using the TissueLyzer LT bead mill (Qiagen) for 4 min at a frequency of 50/s. Total protein amount was measured by BCA assay (ThermoFisher Scientific). After SDS–PAGE and protein transfer onto nitrocellulose membranes (GE Healthcare), probing of specific proteins was performed using indicated primary antibodies and horseradish peroxidase-conjugated donkey anti-rabbit and sheep anti-mouse antibodies (Sigma-Aldrich). Monoclonal or polyclonal antibodies against ADAM8 (1:1000, 23778-1-AP, Proteintech), ADAM10 (1:1000, ab124695, Abcam), ADAM10 (1:1000, AB19026, Millipore), ADAM12 (1:1000, 14139-1-AP, Proteintech), ADAM17 (1:1000, 24620-1-

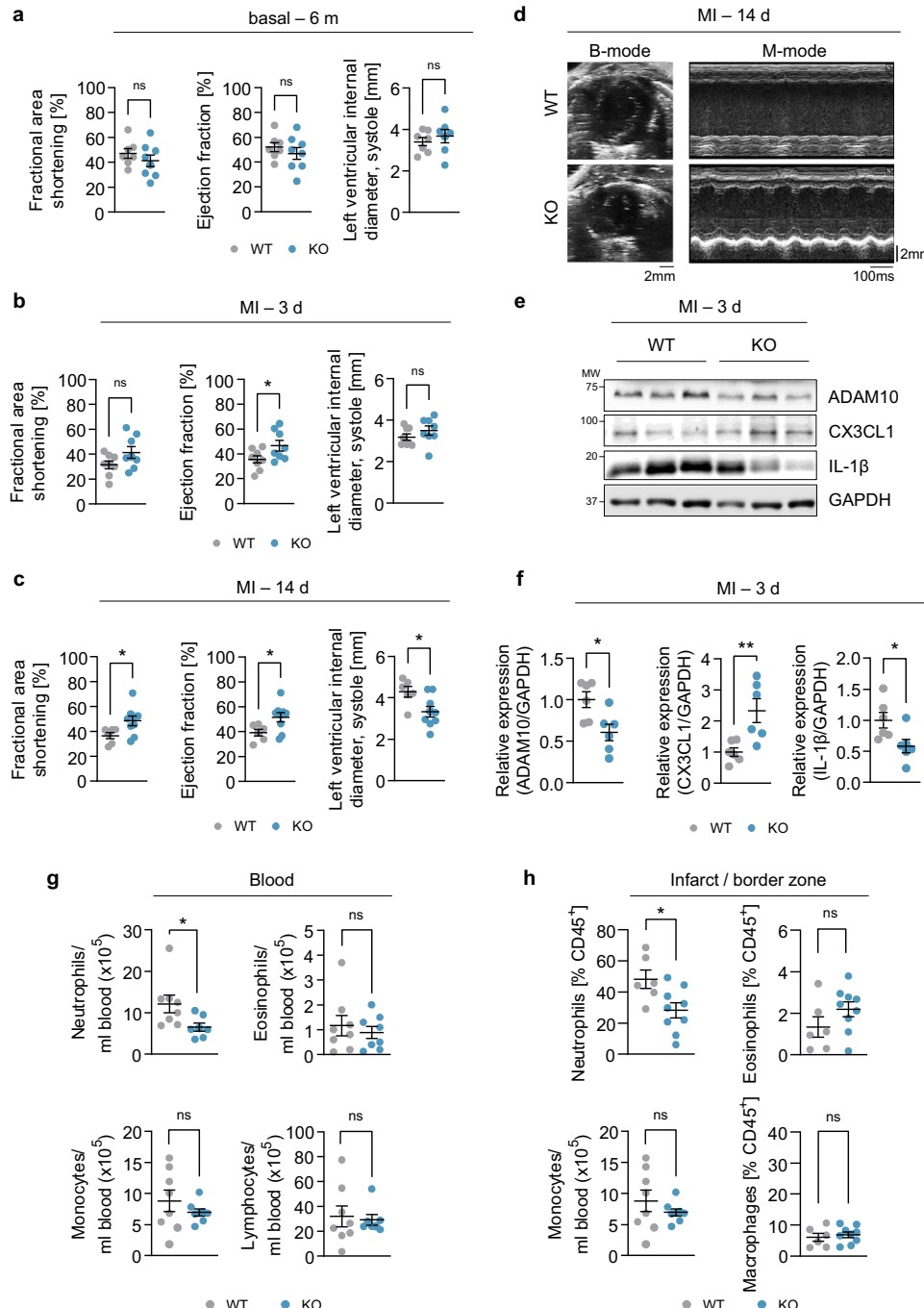

**Fig. 6 | Cardiomyocyte-specific ADAM10 KO improves post-MI cardiac function. a–c** Echocardiographic assessment of fractional area shortening, ejection fraction and left ventricular end-systolic interior diameter in 8-week-old ADAM10$^{fl/fl}$ (WT) and (αMHC-Cre) ADAM10 KO mice under **a** basal conditions (WT, $n = 7$, KO, $n = 8$, mean ± SEM, ns not significant) as well as **b** 3 days (WT, $n = 8$, KO, $n = 9$, mean ± SEM, ns not significant, *$P < 0.05$, two-tailed $t$ test) and **c** 14 days after myocardial infarction (WT, $n = 6$, KO, $n = 9$, mean ± SEM, ns not significant, *$P < 0.05$, two-tailed $t$ test, exact $P$-values are provided in the Source Data file). **d** Representative end-systolic B-mode and M-mode echocardiograms of ADAM10 WT and ADAM10 KO mice 14 days after myocardial infarction. **e** Western blot analysis and **f** quantification of ADAM10, CX3CL1 and IL-1β expression in heart

tissue lysates of ADAM10 WT and ADAM10 KO mice 3 days after myocardial infarction ($n = 6$, mean ± SEM, ns not significant, *$P < 0.05$, **$P < 0.01$, two-tailed $t$ test). Exact $P$-values: ADAM10, 0.0159. CX3CL1, 0.0084. IL-1β, 0.0317. **g** Analysis of leukocyte counts in blood samples of ADAM10 WT ($n = 8$) and ADAM10 KO ($n = 7$) mice (mean ± SEM, ns not significant, *$P < 0.05$, two-tailed $t$ test). Exact $P$-values: Neutrophils, 0.0404. Eosinophils, 0.7886. Monocytes, 0.3373. Lymphocytes, 0.751. **h** Analysis of leukocyte counts in the infarcted area/ infarct border zone of ADAM10 WT ($n = 8$) and ADAM10 KO ($n = 7$) mice (mean ± SEM, ns not significant, *$P < 0.05$, two-tailed two-tailed $t$ test). Exact $P$-values: Neutrophils, 0.0214. Eosinophils, 0.176. Monocytes, 0.639. Macrophages, 0.6064. Source data are provided as a Source Data file.

---

AP, Proteintech), ADAM19 (1:1000, NBP1-69364, Novus biologicals), CXCL16 (1:1000, MAB503, R&D Systems), CX3CL1 (1:1000, AF472, R&D Systems), Fas-L (1:1000, sc-957, Sant Cruz), GAPDH (1:5000, sc-365062, Santa Cruz), IL-1β (1:1000, #12242, Cell Signaling), N-Cadherin

(1:1000, 610921, BD Biosciences), Notch1 (1:1000, ab8925, Abcam), TNFα (1:500, #11948, Cell Signaling) were used. Proteins were visualized using the ECL substrate (SuperSignalTM West Femto Maximum Sensitivity Substrate, resp. SuperSignalTM West Dura Extended

Duration Substrate, ThermoFisher Scientific) and images were acquired using a Fusion FX chemiluminescence imaging system (Vilber). Densitometric analysis was performed using the FusionCapt Advance software (Vilber).

## Immunofluorescence
To analyze the endogenous localization of ADAM10 (1:100, AB19026, Millipore) and CX3CL1 (1:100, ab25088, Abcam), heart tissue sections (mid-ventricular short axis sections, 4% PFA fixed, paraffin embedded, 3 μm) and indicated cells were stained with specific antibodies and counterstained with CD31 (1:100, 553370, BD Biosciences), CD45 (1:100, 555480, BD Biosciences), cTnI (1:100, MAB3150, Millipore), cTnT (1:100, ab45932, Abcam) or Vimentin (1:100, 550513, BD Biosciences) where indicated. Alexa Fluor 488-coupled goat anti-rabbit (1:200, A27034, ThermoFisher Scientific), Alexa Fluor 546-coupled goat anti-mouse (1:200, A-11003, ThermoFisher Scientific) or CF 594-coupled goat anti-rat (1:200, SAB4600323, Sigma-Aldrich) and ProLong Diamant Antifade Mountant with DAPI (P36962, ThermoFisher Scientific) were used as secondary antibodies and for mounting, respectively. Image acquisition was performed using a Keyence BZ-X710 All-in-One Fluorescence Microscope (Keyence) with the BZ-X Viewer v1.03.00.05 software (Keyence). The cellular distribution of ADAM10 and CX3CL1 was assessed using CellProfiler 4[46]. Cardiac-TnT, CD31, CD45 and Vimentin positive cells were analyzed for ADAM10 expression. For each mouse at least 1000 cells/staining were analyzed. Cardiomyocyte cross-sectional diameter was determined using Fiji. One hundred cells/mouse were analyzed.

## Analysis of publicly available single cell sequencing data
The annotated human single cell gene expression dataset "Heart Global" was retrieved from the Heart Cell Atlas (https://cellgeni.cog.sanger.ac.uk/heartcellatlas/data/global_raw.h5ad). The data were analyzed with Scanpy 1.8.2 and the complete software stack is available in a singularity container (https://gitlab.hrz.tu-chemnitz.de/dcgc-bfx/singularity/singularity-single-cell, tag e67259e). First, cells were annotated with the cell types. NotAssigned or doublets were removed. Next, the expression data was normalized to counts per million using the scanpy function 'pp.normalize_total'. Then, the expression data were log10-transformed using the scanpy function 'pp.log1p'. Finally, UMAP- and dot-plots were created using the scanpy functions 'pl.umap' and 'pl.dotplot', respectively.

## Myocardial infarction and drug treatment
At the age of 8 weeks (C57BL/6J) or 8–12 weeks (ADAM10 WT/KO), the left anterior descending coronary artery of mice was permanently ligated to induce MI. After anesthesia with a combination of medetomidine (0.5 mg/kg), midazolam (5 mg/kg) and fentanyl (0.05 mg/kg) (i.p.), the chest was opened between the third and the fourth rib. The left anterior descending coronary artery was occluded using an 8–0 silk suture. Anesthesia was antagonized using a combination of atipamezole (0.75 mg/kg), flumazenil (0.5 mg/kg) and naloxone (1.2 mg/kg) (i.p.). Where indicated, osmotic minipumps (ALZET 1003D/ALZET 2002; DURECT) containing either DMSO or GI254023X (100 mg/kg dissolved in DMSO, Endotherm) were subcutaneously implanted for continuous dosing. For postoperative analgesia, animals received metamizole (300 mg/kg, p.o.). After surgery, the animals were continuously observed until awakening. On the two following days, the mice were assessed daily, and subsequently once a week. Echocardiography was performed under isoflurane anesthesia (0.5 – 4% in oxygen). After final echocardiography, animals were sacrificed painlessly (still under isoflurane anesthesia) by an overdose of thiopental (400 mg/kg, i.p.). This was followed by blood collection via puncture of the heart and subsequent heart removal.

## Assessment of cardiac function
Echocardiography was performed using a Vevo 3100 system (VisualSonics) as described previously[47]. In brief, animals were anesthetized (2% v/v isoflurane) and kept under surveillance. Surface ECG obtained using limb electrodes and body temperature (37 °C warming plate, anal probe) were closely monitored. The mice were shaved on the chest using depilatory cream and held fixed on the warmed mounting plate in prone position. Warm ultrasound coupling gel (37 °C) was placed on the shaved chest area, and the MX400 transducer was positioned to obtain 2D B-mode parasternal long and short axis views and a M-mode view. Echocardiographic analysis was performed using the Vevo 2.1.0 software (VisualSonics). Fractional area shortening (FAS, %) was calculated from left ventricular endocardial area in end-diastole and end-systole. Ejection fraction (EF, %) was calculated from left ventricle length at end-diastole and end-systole and left ventricular short axis area in end-diastole and end-systole.

## Immunohistochemistry
For histological analysis, mid-ventricular short axis sections were fixed in 4% PFA for 24 h before dehydration using an ethanol gradient. Subsequently the samples were embedded in paraffin, sectioned at 3 μm thickness and stained with sirius red (Sigma-Aldrich). Image acquisition was performed using the Keyence BZ-X710 All-in-One Fluorescence Microscope (Keyence) with the BZ-X Viewer v1.03.00.05 software (Keyence). Using Fiji, the epicardial and endocardial infarct length and circumference was measured. Infarct size was calculated using the following equation: [(epicardial infarct length/epicardial circumference) + (endocardial infarct length/endocardial circumference)/2] * 100[47].

## RNA isolation and quantitative real-time polymerase chain reaction (qPCR)
Ventricular myocardium (30 mg) was collected in Qiazol (Qiagen) and homogenized using the TissueLyzer bead mill (Qiagen) for 4 min at a frequency of 50/s. For RNA isolation and subsequent cDNA synthesis the miRNAeasy kit including DNAse1 digestion (Qiagen) was used according to the manufacturer's instructions. RNA (1 μg) was reverse-transcribed into cDNA using the iScript cDNA Synthesis kit (BioRad) following the manufacturer's instructions. Quantitative real-time PCR (RT-qPCR) was performed using the CFX96 C1000 Touch™ thermal cycler Real-Time System (BioRad) in combination with the SsoAdvanced™ Universal SYBR® Green Supermix (BioRad) according to the manufacturer's instructions. Samples were amplified in duplicates and gene expression was quantified using the CFX manager 3.1 software (BioRad) applying the $2^{-\Delta\Delta Ct}$ method[48]. Relative gene expression was calculated to the housekeeping gene *Hprt1*. Specific primer sequences are provided in Supplementary Table 1.

## RNA sequencing
mRNA was isolated from 300 ng total RNA by poly-dT enrichment using the NEBNext Poly(A) mRNA Magnetic Isolation Module (NEB) according to the manufacturer's instructions. Samples were then directly subjected to the workflow for strand-specific RNA-Seq library preparation (Ultra II Directional RNA Library Prep, NEB). For ligation, custom adapters were used (Adapter-Oligo 1: 5′-ACACTCTTTCCCTACACGACGCTCTTCCGATCT-3′, Adapter-Oligo 2: 5′-P-GATCGGAAGAGCACACGTCTGAACTCCAGTCAC-3′). After ligation, adapters were depleted by an XP bead purification (Beckman Coulter) adding the beads solution in a ratio of 1:0.9. Dual indexing was done during the following PCR enrichment (12 cycles, 65 °C) using custom amplification primers carrying the same sequence for i7 and i5 index (Primer 1: AATGATACGGCGACCACCGAGATCTACACNNNNNNNNNACATCTTTCCCTACACGACGCTCTTCCGATCT, Primer 2: CAAGCAGAAGACGGCATACGAGATNNNNNNNNNGTGACTGGAGTTCAGACGTGTGCTCTTCCGATCT). After 2x XP bead purification step (1:0.9), libraries were

quantified using the Fragment Analyzer (Agilent). For Illumina flow-cell production, libraries were equimolarly pooled and sequenced with 100 bp paired-end on an Illumina NovaSeq 6000, averaging 32 million fragments per library. After sequencing, RNA-SeQC (1.1.8)[49] was used to perform a basic quality control which includes exonic, intronic and intergenic distribution of the reads and rRNA rate within each sample. Alignment of the reads to the mus musculus reference (mm10, RefSeq assembly accession code: GCF_000001635.20) was done with GSNAP (v2020-12-16)[50], and Ensembl gene annotation version 98 was used to detect splice sites. The uniquely aligned reads were counted with featureCounts (v2.0.1)[51] and the same Ensembl annotation. Normalization of the raw fragments counts based on the library size and testing for differential expression between the conditions was performed with the R packages (v.4.0.4) DESeq2 (v1.30.1)[52] and IHW (1.18.0)[53]. Genes with an adjusted $p$-value (padj) < 0.01 were considered as differentially expressed. Gene ontology and Gene Set Enrichment Analysis (GSEA) were performed using Enrichr (version March2021)[54] and the GSEA tool (v4.1.0) of the Broad Institute[55]. Here, a GSEA PreRanked was performed where the stat column (Wald statistic) of DESeq2 test for differential gene expression was used as input. Mouse Ensembl IDs were internally converted to Human IDs. RNA sequencing data have been deposited in the Gene Expression Omnibus (GEO) under accession code GSE217268.

## Enzyme-linked immunoassays

Enzyme-linked immunoassays (ELISA) to determine CXCL1 (ab216951, Abcam), CXCL5 (ab264611, Abcam), CXCL16 (ab197744, Abcam), CX3CL1 (ab100683, Abcam), IL-1β (ab197742, Abcam) and IL-6 (KE10007, Proteintech) levels in serum or cell culture supernatant were conducted according to the manufacturer's instructions. Absorption was analyzed using the BioTek Synergy HTX reader (Bio-Tek) with the Gene5 v2.09 software (BioTek).

## Flow cytometry

Single-cell suspensions were obtained from peripheral blood and heart tissue. Blood was collected by cardiac puncture in 50 mmol/L EDTA, and subsequently analyzed for leukocyte counts using fluorescence flow cytometry with the Sysmex XN analyzer (Sysmex). For analysis of leukocyte heart tissue infiltration, mouse hearts were perfused through the left ventricle with 5 mL ice-cold PBS, the infarct area dissected and minced with a fine scissor and digested in RPMI 1640 medium containing 1 mg/mL collagenase I (Roche) and 0.1 mg/mL DNAse I (Roche) twice for 20 min at 37 °C under agitation. Tissues were triturated and cells filtered through a 50-μm filter (CellTrics, Sysmex), washed, and centrifuged (5 min; 500 g; 4 °C). Subsequently, cells were stained with Fc blocking antibody (1:200) for 10 min on ice. Following Fc block, cells were stained for 30 min at 4 °C in PBS with FACS buffer (PBS supplemented with 2% FBS and 2 mM EDTA). Fluorochrome-conjugated antibodies specific to mouse CD45 (1:75, 103147, Biolegend), CD11b (1:100, 12-0112-82, ThermoFisher Scientific), Ly6G (1:100, 127624, BioLegend), Siglec-F (1:100, 562680, BD Bioscience), F4/80 (1:100, 25-4801-82, ThermoFisher Scientific) and Ly6C (1:100, 553104, BD Biosciences) were used. Neutrophils were identified as CD45$^+$, CD11b$^+$, Ly6G$^+$. Eosinophils were identified as CD45$^+$, CD11b$^+$, Ly6G$^-$, SigF$^+$. Monocytes were identified as CD45$^+$, CD11b$^+$, Ly6G$^-$, SigF$^-$, F4/80$^-$, Ly6C$^+$. Macrophages were identified as CD45$^+$, CD11b$^+$, Ly6G$^-$, SigF$^-$, F4/80$^+$. Dead cells were identified through DAPI (ThermoFisher Scientific) staining directly before FACS analysis. Data were acquired on an LSR Fortessa X-20 (BD Biosciences) with the BDFACS Diva software version 8.0.2 (BD Biosciences) and analyzed with the FlowJo version 10.8.0 software (BD Biosciences).

## Co-Immunoprecipitation

For analysis of endogenous protein interactions, 100-mm dishes of HL-1 cells were used to generate whole-cell lysates. Cells were harvested using cell lysis buffer (Cell Signaling) and cOmplete protease inhibitor cocktail (Roche). Total protein amount was measured by BCA assay (Thermo Fisher Scientific). Cell lysates were pre-cleared using 50 μl of a protein AG sepharose slurry (50% v/v) (Alpha Diagnostics). Primary antibodies (ADAM10, ab124695, Abcam) were added to the 0.5 mg protein lysate and rotated for 1 h at 4 °C. Subsequently, 50 μl of a protein AG sepharose slurry (50% v/v) was added and rotated overnight at 4 °C. Co-Immunoprecipitates were washed 3X with cold lysis buffer. Whole-cell lysates and co-precipitated proteins were boiled in 50 μl sample buffer, separated by SDS-PAGE, transferred, and blotted. Protein precipitates were analyzed with specific primary antibodies as indicated.

## Isolation of bone marrow derived cells

To isolate whole bone marrow cells, long bones of 8–10-week-old C57BL/6J mice were triturated with mortar and pestle in PBS supplemented with 0.5% bovine serum albumin and 2 mM EDTA. Cells were filtered through a 70-μm filter (Miltenyi Biotech), washed, and centrifuged (10 min; 300 g; 4 °C). After erythrocyte lysis in hypotonic NH$_4$Cl − buffer, cells were filtered through a 30-μm filter (Miltenyi Biotech). For neutrophil isolation the mouse Neutrophil Isolation Kit (130-097-658, Miltenyi Biotec) was used according to the manufacturer's instructions. Bone marrow derived non-neutrophils that were retained in the separator column after neutrophil isolation were plated in non-tissue culture treated dishes and treated with 20 ng/μl murine M-CSF (315-02, Peprotech) for 7 days to generate macrophages[56].

## Small interfering RNA knockdown and inhibitor treatment

Specific siRNAs against murine CX3CL1 (SR426672A, SR426672C) and unspecific control siRNA (SR300004) were obtained from Origene. Twenty-four hours after cell plating, 20 nM siRNA was delivered using Lipofectamine RNAiMAX (ThermoFisher Scientific). Forty-eight hours after transfection, cells were subjected to normoxic and hypoxic (1% O$_2$) conditioning or lysed for Western Blot analysis. For pharmacological inhibition of ADAM10 or simulation of hypoxia, cells were treated with GI254023X (10 μM, Endotherm) or DMOG (1 mM, Sigma-Aldrich) for 24 h, respectively. DMSO was used as control.

## Hypoxic conditioning and transwell migration assay

To generate chemoattracting media for in vitro migration assays, HL-1 cells were plated in 0.5% fibronectin, 0.02% gelatin (all Sigma-Aldrich) coated 6-well plates. After 4 d of incubation at 37 °C in a humidified atmosphere containing 5% CO$_2$, cells were treated with eighter DMSO or GI254023X for 24 h. Subsequently, cells were washed with fresh medium and subjected to 3 h of normoxic and hypoxic (1% O$_2$) conditioning. Cells were lysed for Western Blot analysis and supernatants were used as chemoattractant in the lower chamber of 24-well Transwell plates with 8-μm pore size (ThermoFisher Scientific). Following isolation (neutrophils) or differentiation (macrophages), bone marrow derived cells were suspended in RPMI without FBS, plated in each upper chamber of the Transwell plates and incubated for 12 h at 37 °C in a humidified atmosphere containing 5% CO$_2$. Following incubation, cells were fixed with 4% PFA for 15 min, DAPI stained and mounted on microscopy slides using ProLong Diamant Antifade Mountant (ThermoFisher Scientific). Image acquisition was performed using the Keyence BZ-X710 All-in-One Fluorescence Microscope (Keyence). Cell counts per field were analyzed using CellProfiler 4.0[46].

## Statistics

Data are presented as means ± SEM. GraphPad Prism version 9.4.1 software (Dotmatics) was used for statistical analysis. Kolmogorov−Smirnov normality test was performed in all experimental groups to test for Gaussian distribution. If the data was normally distributed, one-way ANOVA followed by Tukey's test (multiple

comparisons between different groups) or two-tailed unpaired Student's *t* tests (comparison between two groups) were performed. If the data was not normally distributed, Kruskal–Wallis test followed by Dunn's multiple comparison test (multiple comparisons between different groups) and Mann–Whitney test (comparison between two groups) were performed. Statistical analysis is indicated in the respective figure legends. $P < 0.05$ was considered statistically significant.

## Reporting summary

Further information on research design is available in the Nature Portfolio Reporting Summary linked to this article.

## Data availability

The annotated human single cell gene expression data to evaluate ADAM10 expression were retrieved from the Heart Cell Atlas [https://cellgeni.cog.sanger.ac.uk/heartcellatlas/data/global_raw.h5ad]. The mus musculus reference genome mm10 (RefSeq assembly accession code: GCF_000001635.20) was used to align RNA sequencing reads. The RNA sequencing data generated in this study have been deposited in the Gene Expression Omnibus (GEO) under accession code GSE217268. Source data are provided with this paper.

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

## Acknowledgements
We are grateful to Nadine Weser, Manja Newe, Annett Opitz, Romy Kempe and Konstanze Fischer for their excellent technical assistance. This work was supported by the German Research Foundation (DFG) grants EL 270/7-3, Transregio-SFB CRC/TRR 205 to A.E.A., grant KA 4194/3-3 to S.K., project no. 288034826 - international research training group (IRTG) 2251 to A.E.A., K.G. and S.K., grant WA 2586/4-1 to M.W. and SFB877-A3 to P.S., SFB1525-B03/453989101 and CRC/TRR 296/1-P10/ 424957847 to K.L., by research stipend of the German Heart Foundation/ German Foundation for Heart Research to S.K., and F/34/19 to E.K., and of the German Cardiac Society DGK10/2021 to E.K. and the Foundation for Pathobiochemistry and Molecular Diagnostics, no. 060.1052 to P.M.

## Author contributions
Conceptualization: E.K., A.E.A. Methodology: E.K., S.W., P.M., A.D., A.W., I.K., K.L., K.G., A.E.A. Investigation: E.K., A.W., D.S., P.K., J.W., N.V., S.R.K., M.G., S.W., P.M., M.L., F.R., A.D., K.L. Resources: P.S., E.R., L.K., A.E.A. Visualization: E.K., M.W., A.E.A. Funding acquisition: E.K., S.K., K.G., M.W., A.E.A. Project administration: E.K., A.E.A. Supervision: E.K., A.E.A. Writing – original draft: E.K., M.W., A.E.A. Writing – review & editing: E.K., M.W., P.S., A.E.A.

## Funding

## Competing interests
The authors declare no competing interests.
