## [Peer Review File · Nature Communications]

Targeting cardiomyocyte ADAM10 ectodomain shedding promotes survival early after myocardial infarctionREVIEWER COMMENTS

Reviewer #1 (Remarks to the Author):

This is an interesting study regarding a novel role of ADAM10 ectodomain shedding of cytokine CX3CL1 to augment neutrophil accumulation and associated myocardial injury after an infarction. The data presented are rather comprehensive, with in vivo justification in both mouse models and relevant human disease, dissection of the pathways with appropriate loss of function studies and new genetic models. The results are well presented, clear and convincing. The study is really the first to identify this signaling complex and potential target to ameliorate the post MI condition.

I really have only two main comments and two suggestions.

The models presented - both pharmacological inhibition study using A10i, and myocyte selective KO of ADAM10 - block the pathway prior to the infarction insult to the heart. However, it is unlikely that ADAM10 interference therapy would be administered in this manner. At best it would be at time of presentation is typically hours if not a day or more after the infarction onset. As the study concludes the strategy works by targeting neutrophils and their adverse influence on the myocardium, this suggests its efficacy requires very early administration, and that if administered say 3 days post MI, would have little impact. I see value to testing both. One would be administration after day 3 as this could further support that neutrophil accumulation is indeed the key target of ADAM10 suppression. I would then also test administration say 6 hours after MI - a practical window for clinical therapy but still well before peak neutrophil accumulation in the heart. Retained efficacy with this strategy would add important translational significance to the findings. These studies would not take that long to accomplish given the MI model is relatively short (2 weeks), and the drug available. I think they would add to the strength of an already fine study.

My only other comment is that the stats are generally reported as using a one way ANOVA or T-tests. However, often there are rather broad differences in the variance among the groups, e.g. Figure 3H and 3I - with plenty of group data showing non-normal distributions. Thus, non-parametric analysis, or Welch-ANOVA and T-tests where uniform variance is not assumed are appropriate there. This needs to be carefully reviewed throughout.

Reviewer #2 (Remarks to the Author):

This manuscript addresses the potential role of the cell surface metalloprotease ADAM10 in cardiomyocyte survival after myocardial infarction (MI) in mice. Most of the experiments are focused on the effects of an ADAM10-selective inhibitor (GI254023X (A10i)) in a mouse model of MI, which is triggered by ligation of the left anterior descending artery (LAD). The authors provide evidence for an upregulation of ADAM10 in the infarcted area in this model and in patients suffering from ischemic cardiomyopathy. Treatment of mice subjected to the LAD model with A10i improves their survival, heart function and heart tissue repair. Transcriptome analysis of treated and untreated animals showed that the A10i blocks neutrophil recruitment, possibly by inhibiting the shedding of fractalkine (CX3CL1), a membrane-anchored protein with an important role in recruitment of immune cells such as neutrophils. This conclusion was further supported by ex vivo experiments, in which iPSC-derived cardiomyocytes were co-cultured with bone marrow-derived neutrophils and macrophages. Under these conditions, the migration of neutrophils, but not macrophages, could be blocked by A10i and by siRNA against fractalkine (CX3CL1). To further corroborate that ADAM10 in cardiomyocytes could be the target of the A10i, the authors generated conditional knockout mice to remove ADAM10 from cardiomyocytes and showed that these animals had better heart function than controls subjected to the LAD MI model. Based on these experiments, the authors conclude that ADAM10 in myocardial cells has a role in exacerbating the consequences of MI by releasing Fractalkine to recruit neutrophils, suggesting that ADAM10 would be a good target for treatment of MI in patients.

Critique:

The main concern about this manuscript is that most of the mechanistic studies were performed with the A10i, which is somewhat selective for ADAM10, but is also known to inhibit other metalloproteases, including the related ADAM17. Moreover, ADAM10 has many substrates and has a crucial role in Notch signaling, yet the discussion is mainly focused on one possible substrate, CX3CL1, even though other pathways could also be affected by the A10i, including IL-1b (see figure 3). Importantly, the experiments that could most directly address the role of cardiomyocyte ADAM10 in the LAD MI model in conditional ADAM10 knockout mice do not include several of the most important mechanistic experiments that were only performed with mice treated with the A10i. Thus, while the experiments presented in this study are consistent with the authors conclusions that blocking ADAM10 could improve the outcome of an MI by preventing the release of CX3CL1, there are many other possible interpretations that are not sufficiently considered in the discussion. In this reviewer's opinion, the authors would have to provide much stronger evidence in support of the proposed crucial role of an ADAM10/CX3CL1 axis in MI to justify publication of this study in Nature Communications.

Here are some more specific comments for the authors:

- 1) In Figure 1, the molecular weight of ADAM8 and ADAM17 does not appear to be correct. Please provide more details on the lysis conditions for these Western blots, also for ADAM10. Please note that both ADAM10 and ADAM17 require specific lysis conditions and inclusion of metalloprotease inhibitors to accurately detect the mature form (see McIlwain et al., Science. 2012 Jan 13;335(6065):229-32 for ADAM17 and Brummer, T et al. FASEB J. 2018 Jul;32(7):3560-3573 for ADAM10). Western Blots for ADAM10 and ADAM17 should be performed under conditions where the authors can rule out the autodegradation of the mature form of ADAM10 or ADAM17 described in these manuscripts.
- 2) The authors should provide evidence that 100mg/kg of the A10i is indeed a selective concentration for ADAM10 over ADAM17, for example by testing how this concentration affects the release of TNF α from LPS-stimulated BMDM, which is mainly dependent on ADAM17. In general, the authors should discuss the effects of the A10i instead of the "effect of ADAM10 inhibition", since this is only a partially selective inhibitor that can also block other ADAMs and MMPs.
- 3) The immunofluorescence analysis in Figure 2 does not provide rigorous evidence for the activation state of ADAM10, and the cited JBC paper uses mainly a chimera between the ADAM10 cytoplasmic domain and Tac. It is also not clear how a hydroxamate type inhibitor such as the A10i should affect the maturation or subcellular localization of ADAM10? Figure 2h and the corresponding results section should be removed, as it is not clear that these experiments support the authors interpretation.
- 4) In this reviewer's opinion, the discussion is too focused on ADAM10/CX3CL1 signaling as the main target of the A10i and a main pathogenic mechanism in MI. It is also possible that the effect of the A10i on other processes, such as shedding of other substrates of ADAM10 or ADAM17 or inhibition of other ADAMs or MMPs, offer the described protection from MI. The blockade of IL-1b is a case in point.
- 5) The cardiomyocytes used to determine the expression of ADAM10 were derived from iPSC, so it would be good to independently verify the statement that "cardiac ADAM10 is mostly expressed in cardiomyocytes" (top of page 13) using cardiomyocytes isolated from mouse hearts, or other approaches, such as immunofluorescence analysis or expression analysis (scSEQ or qPCR from FACS-sorted or bead isolated heart cell types) to support this conclusion.
- 6) Regarding Figure 5 b-d, if CX3CL1 is shed by ADAM10, shouldn't there be more in the presence of the A10i inhibitor? Moreover, it is not clear how and why an A10 inhibitor would block the co-IP of the enzyme and substrate? And finally, the IF data in Figure 5d are also difficult to interpret.
- 7) Regarding the Western blots in Supplementary Figure 5A, there is large variability in the p-PLB samples at 6 m. How meaningful are these data? The actual blots for the MI samples at 14d are not much more convincing, with such large variability between samples that it is difficult to draw conclusions about the biological significance of these findings
- 8) Many key findings were not addressed in the conditional cardiomyocyte ADAM10 ko mice, such as CX3CL1 shedding and neutrophil recruitment.
- 9) Re the model shown in Supplementary Figure 6, there is no meaningful discussion of the

transcriptional regulation of the various genes that are affected by the A10i. Only CX3CL1 shedding is mentioned, although there are many other possible mechanisms for how the A10i could affect the outcome of the LAD MI model.

Reviewer #3 (Remarks to the Author):

This study provides evidence that pharmacological inhibition and myocyte-specific deletion of ADAM10 improves cardiac remodeling after myocardial infarction. This goes along with diminished CX3CL1 ectodomain shedding and reduced neutrophil infiltration into the injured heart. There are, however, some conceptual and experimental shortcomings which finally question the magnitude and biological relevance of the observed effect and the cellular subpopulation involved in the infarcted heart.

Major

- a. The assignment and cellular distribution of ADAM10 is not convincing. The authors have used iPS-derived CM and a mouse cell line resembling features of CM but not CM freshly isolated from control and infarcted hearts. Similarly, the fibroblasts used are likely not to resemble the activated fibroblasts present in the infarcted heart. Therefore it is quite possible that aside of ADAM10 in cardiomyocytes other non-cardiomyocytes contribute to the ADAM10 signal in the infarcted heart. What is needed is a direct comparison of freshly isolated mouse CM with FACS- sorted non-CM (endothelium, fibroblasts, immune cells). Helpful in this regard would also be a search on the cellular distribution of expressed ADAM10 in published single cell sequencing data sets. This in part may resolve the question of cellular specificity. Also note that the recruitment of blood leukocytes across the endothelium to sites of tissue inflammation was reported to involve disintegrin and ADAM10 that was implicated in leukocyte transmigration by proteolytically cleaving its endothelial substrates (DOI: 10.4049/jimmunol.1600713)
- b. Genetic deletion of ADAM10 is experimentally certainly superior to using systemic application of the experimental ADAM10 inhibitor GI254023X, because of the absence of possible pharmacologic side effects. In the genetic model the improvement of functional parameters (figure 6 e) was only rather small as compared with the massive improvement of ejection fraction observed after treatment with GI254023X. This strongly argues that pharmacologic inhibition has side effects and/or may target other cell types within the heart aside of cardiomyocytes (see above). As to the first possibility there was a clinical trial with GI254023X which was discontinued in phase I/II due to hepatotoxicity following systemic administration (doi: 10.3389/fimmu.2020.00499).
- c. The authors should consider to reconstruct the manuscript putting the data with the cardio-specific ADAM10 KO in a central position, showing the human data at the end to point out translational potential.

Minor

- d. It is somewhat disturbing that summary and introduction start with the identical sentence.
- e. The immunofluorescence images shown in figure 2h require co-staining with cellular markers for cardiomyocytes and non-cardiomyocytes to again prove the localization of ADAM10 in cardiomyocytes.
- f. Since the ejection fraction after 28 days in the treated infarct group is not distinguishable from controls while there is still a scar is formed (figure 2 g) this argues that the remaining myocardium most likely is hypertrophied to compensate for the lost cardiomyocytes.
- g. Data shown in figure 3 are derived from transcriptome analysis of the entire infarcted area. As was discussed above a single cell transcriptomics analysis should have been performed.
- h. Figure 6 b , c are basal controls and should go into the appendix

Responses to Reviewers

Reviewer #1 (Remarks to the Author):

This is an interesting study regarding a novel role of ADAM10 ectodomain shedding of cytokine CX3CL1 to augment neutrophil accumulation and associated myocardial injury after an infarction. The data presented are rather comprehensive, with in vivo justification in both mouse models and relevant human disease, dissection of the pathways with appropriate loss of function studies and new genetic models. The results are well presented, clear and convincing. The study is really the first to identify this signaling complex and potential target to ameliorate the post MI condition.

We thank the Reviewer for critically reviewing our manuscript and the helpful comments and suggestions.

I really have only two main comments and two suggestions.

The models presented - both pharmacological inhibition study using A10i, and myocyte selective KO of ADAM10 - block the pathway prior to the infarction insult to the heart. However, it is unlikely that ADAM10 interference therapy would be administered in this manner. At best it would be at time of presentation is typically hours if not a day or more after the infarction onset. As the study concludes the strategy works by targeting neutrophils and their adverse influence on the myocardium, this suggests its efficacy requires very early administration, and that if administered say 3 days post MI, would have little impact. I see value to testing both. One would be administration after day 3 as this could further support that neutrophil accumulation is indeed the key target of ADAM10 suppression. I would then also test administration say 6 hours after MI - a practical window for clinical therapy but still well before peak neutrophil accumulation in the heart. Retained efficacy with this strategy would add important translational significance to the findings. These studies would not take that long to accomplish given the MI model is relatively short (2 weeks), and the drug available. I think they would add to the strength of an already fine study.

We agree that the translational aspect of our study can be further strengthened by an additional experimental study design. Regarding the first suggestion, starting therapy after three to six hours, we have already done this in a sense. It takes in fact 4-6 hours for the osmotic pumps to reach their nominal transport rate, because these pumps need time to soak up and reach body temperature (<https://www.alzet.com/guide-to-use/filling-priming-alzet-pumps>). Therefore, treatment onset is indeed close to the timepoint suggested by the Reviewer. To emphasize this point we have added the following sentence to the corresponding results section:

Pumps were not primed before implantation, which leads to a delayed onset of substance release (4-6 hours) mimicking the potential clinical scenario of treatment after onset of MI.

Regarding the second suggestion, the reviewer recommends waiting for three days before starting therapy to further support that neutrophil accumulation is indeed the key target of ADAM10 suppression. In this setting, the reviewer argues that if we saw no effect this would provide evidence that neutrophil infiltration is mechanistically decisive for the better outcome. We fully agree. However, we decided to do it the other way around and to stop therapy after three days instead of starting therapy after three days. A positive outcome in this setting would allow to draw the same conclusion and from our point of view in a more translational / therapeutic way. Indeed, we show now that application of the ADAM10 inhibitor for three days after MI is sufficient for preservation of pump function 14 and 28 days after MI (**Fig. 4 d-f of the revised manuscript, see below**). This is a crucial experiment to support our overall conclusion that neutrophil infiltration is indeed the key target of A10i treatment. The corresponding text in the Results section reads as follows:

Treatment with A10i for 3 days after MI is as effective as treatment for 14 days

The fact that the effects of MI on CX3CL1, neutrophil infiltration, IL-1 β and pump function were visible as soon as day three post MI led us to the hypothesis that treatment with A10i for 3 days after MI may be sufficient for the therapeutic effect. Consequently, we compared cardiac function in mice treated for 3 or 14 days at day 14 and 28 after MI (Fig. 4d). At both time points, cardiac function in animals treated for 3 days was significantly improved and virtually indistinguishable from the animals treated for 14 days (Fig. 4e, f). This opens the opportunity for short-term A10i treatment early after MI to limit infarction size.

Fig. 4. ADAM10 inhibition reduces neutrophil bone marrow egress and heart tissue infiltration following myocardial infarction. **a** Gating strategy for identification of neutrophils (CD45⁺, CD11b⁺, Ly6G⁺), eosinophils (CD45⁺, CD11b⁺, Ly6G⁻, SigF⁺), monocytes (CD45⁺, CD11b⁺, Ly6G⁻, SigF⁻, F4/80⁺, Ly6C⁺) and macrophages (CD45⁺, CD11b⁺, Ly6G⁻, SigF⁻, F4/80⁺) in the infarcted area/ infarct border zone of A10i (n = 5) and DMSO (n = 5) treated mice 3 days after infarction **b** Analysis of leukocyte counts in blood samples of GI254023X (A10i, n = 5) and DMSO (n = 5) treated mice 3 days after infarction (mean \pm SEM, ns = not significant, **P<0.01, Mann-Whitney test). **c** Analysis of leukocyte counts in the infarcted area/ infarct border zone of GI254023X (A10i, n = 5) and DMSO (n = 5) treated mice 3 days after infarction (mean \pm SEM, ns = not significant, *P<0.05, Mann-Whitney test). **d** Treatment scheme. Following LAD ligation (MI), mice were treated with GI254023X (A10i, 100 mg/kg) for 3 days (n = 12), 14 days (n = 7) or DMSO (n = 10) using osmotic minipumps. Final echocardiography and sample collection was performed 14 and 28 days after LAD ligation. **e-f** Echocardiography assessment of fractional area shortening, ejection fraction and left ventricular end-systolic interior diameter **e** 14 and **f** 28 days (DMSO, n = 6, A10i - 3 d, n = 11, A10i - 14 d, n = 7) after myocardial infarction (mean \pm SEM, *P<0.05, **P<0.01, one-way ANOVA with Tukey's posttest).

My only other comment is that the stats are generally reported as using a one way ANOVA or T-tests. However, often there are rather broad differences in the variance among the groups, e.g. Figure 3H and 3I - with plenty of group data showing non-normal distributions. Thus, non-parametric analysis, or Welch-ANOVA and T-tests where uniform variance is not assumed are appropriate there. This needs to be carefully reviewed throughout.

We agree with the Reviewer. We now performed the Kolmogorov-Smirnov normality test in all experimental groups throughout the revised manuscript. If the data was normally distributed, we performed one way ANOVA followed by Tukey's test or Student's t-test when only two groups were compared. If the data was not normally distributed, we used Kruskal-Wallis test followed by Dunn's multiple comparison test and Mann-Whitney test respectively. This altered some p-values, but not the general conclusion from the affected experiments.

Reviewer #2 (Remarks to the Author):

This manuscript addresses the potential role of the cell surface metalloprotease ADAM10 in cardiomyocyte survival after myocardial infarction (MI) in mice. Most of the experiments are focused on the effects of an ADAM10-selective inhibitor (GI254023X (A10i)) in a mouse model of MI, which is triggered by ligation of the left anterior descending artery (LAD). The authors provide evidence for an upregulation of ADAM10 in the infarcted area in this model and in patients suffering from ischemic cardiomyopathy. Treatment of mice subjected to the LAD model with A10i improves their survival, heart function and heart tissue repair. Transcriptome analysis of treated and untreated animals showed that the A10i blocks neutrophil recruitment, possibly by inhibiting the shedding of fractalkine (CX3CL1), a membrane-anchored protein with an important role in recruitment of immune cells such as neutrophils. This conclusion was further supported by ex vivo experiments, in which iPSC-derived cardiomyocytes were co-cultured with bone marrow-derived neutrophils and macrophages. Under these conditions, the migration of neutrophils, but not macrophages, could be blocked by A10i and by siRNA against fractalkine (CX3CL1). To further corroborate that ADAM10 in cardiomyocytes could be the target of the A10i, the authors generated conditional knockout mice to remove ADAM10 from cardiomyocytes and showed that these animals had better heart function than controls subjected to the LAD MI model. Based on these experiments, the authors conclude that ADAM10 in myocardial cells has a role in exacerbating the consequences of MI by releasing Fractalkine to recruit neutrophils, suggesting that ADAM10 would be a good target for treatment of MI in patients.

Critique:

The main concern about this manuscript is that most of the mechanistic studies were performed with the A10i, which is somewhat selective for ADAM10, but is also known to inhibit other metalloproteases, including the related ADAM17. Moreover, ADAM10 has many substrates and has a crucial role in Notch signaling, yet the discussion is mainly focused on one possible substrate, CX3CL1, even though other pathways could also be affected by the A10i, including IL-1b (see figure 3). Importantly, the experiments that could most directly address the role of cardiomyocyte ADAM10 in the LAD MI model in conditional ADAM10 knockout mice do not include several of the most important mechanistic experiments that were only performed with mice treated with the A10i. Thus, while the experiments presented in this study are consistent with the authors conclusions that blocking ADAM10 could improve the outcome of an MI by preventing the release of CX3CL1, there are many other possible interpretations that are not sufficiently considered in the discussion. In this reviewer's opinion, the authors would have to provide much stronger evidence in support of the proposed crucial role of an ADAM10/CX3CL1 axis in MI to justify publication of this study in Nature Communications.

We thank the Reviewer for critically reviewing our manuscript and the helpful comments and suggestions.

According to the suggestions of the Reviewer we have added a completely new series of experiments performed on the cardiomyocyte-specific ADAM10 knockout mice (ADAM10 KO) to the revised manuscript. It addresses the effect of the ADAM10 KO three days after MI. The level of IL-1 β in the infarcted area is lower in ADAM10 KO and the expression of non-shed CX3CL1 is higher (**Fig. 6e, f of the revised manuscript, see below**). Using FACS, we show that like in the case for using the ADAM10 inhibitor, the infiltration of the infarcted myocardium by neutrophils is significantly lower in ADAM10 KO than in WT, as well as the neutrophil number in the blood (**Fig. 6g, h of the revised manuscript, see below**). Moreover, already at three days, pump function is better compared to WT, as was seen in the animals treated with the ADAM10 inhibitor (**Fig. 6b of the revised manuscript, see below**), which is retained at day 14 (**Fig. 6c of the revised manuscript, see below**). The corresponding section in the Results reads as follows:

We next investigated the effects of the ADAM10 KO in MI. As soon as 3 days after LAD ligation, ADAM10 KO mice showed a significantly higher EF compared to WT controls. By day 14, FAS and EF as well as LVID were improved as compared to WT controls (Fig. 6b-d) validating the demonstrated cardioprotective effects of A10i. Importantly, and similar to the results of the A10i, ADAM10 KO resulted in significantly reduced shedding of CX3CL1 in the infarct and border zone and reduced cardiac expression of IL-1 β 3 days after infarction (Fig. 6e, f). ADAM10 expression was significantly reduced.

Also, cardiac neutrophil invasion was significantly lower in ADAM10 KO 3 days after MI as were blood neutrophil counts (Fig. 6 g, h and Supplementary Fig. 7).

Fig. 6. Cardiomyocyte-specific ADAM10 KO improves post-MI cardiac function. **a-c** Echocardiography assessment of fractional area shortening, ejection fraction and left ventricular end-systolic interior diameter in 8-week-old ADAM10^{fl/fl} (WT) and (α MHC-Cre) ADAM10 KO mice under **a** basal conditions (WT, n = 7, KO, n = 8, mean \pm SEM, ns = not significant, *P<0.05, two-tailed t test) as well as **b** 3 (WT, n = 8, KO, n = 9, mean \pm SEM, ns = not significant, *P<0.05, two-tailed t test) and **c** 14 days after myocardial infarction (WT, n = 6, KO, n = 9, mean \pm SEM, ns = not significant, *P<0.05, two-tailed t test). **d** Representative end-systolic B-mode and M-mode echocardiograms of ADAM10 WT and ADAM10 KO mice 14 days after myocardial infarction. **e** Western blot analysis and **f** quantification of ADAM10, CX3CL1 and IL-1 β expression in heart tissue lysates of ADAM10 WT and ADAM10 KO mice 3 days after myocardial infarction (n = 6, mean \pm SEM, ns = not significant, *P<0.05, **P<0.05, two-tailed t test). **g** Analysis of leukocyte counts in blood samples of ADAM10 WT (n = 8) and ADAM10 KO (n = 7) mice (mean \pm SEM, ns = not significant, *P<0.01, Mann-Whitney test). **h** Analysis of leukocyte counts in the infarcted area/ infarct border zone of ADAM10 WT (n = 8) and ADAM10 KO (n = 7) mice (mean \pm SEM, ns = not significant, *P<0.01, Mann-Whitney test).

Supplementary Fig. 7. Cardiomyocyte-specific ADAM10 knockout reduces neutrophil bone marrow egress and heart tissue infiltration following myocardial infarction. Gating strategy for identification of neutrophils (CD45⁺, CD11b⁺, Ly6G⁺), eosinophils (CD45⁺, CD11b⁺, Ly6G⁺, SigF⁺), monocytes (CD45⁺, CD11b⁺, Ly6G⁻, SigF⁺, F4/80⁻, Ly6C⁺) and macrophages (CD45⁺, CD11b⁺, Ly6G⁻, SigF⁺, F4/80⁺) in the infarcted area/ infarct border zone of ADAM10 KO (n = 7) and WT (n = 8) treated mice 3 d after infarction.

Here are some more specific comments for the authors:

1) In Figure 1, the molecular weight of ADAM8 and ADAM17 does not appear to be correct. Please provide more details on the lysis conditions for these Western blots, also for ADAM10. Please note that both ADAM10 and ADAM17 require specific lysis conditions and inclusion of metalloprotease inhibitors to accurately detect the mature form (see McIlwain et al., Science. 2012 Jan 13;335(6065):229-32 for ADAM17 and Brummer, T et al. FASEB J. 2018 Jul;32(7):3560-3573 for ADAM10). Western Blots for ADAM10 and ADAM17 should be performed under conditions where the authors can rule out the autodegradation of the mature form of ADAM10 or ADAM17 described in these manuscripts.

The Reviewer raises a relevant point. Since probing for ADAM8 and 17 results in a number of bands in western blots, the selection of the correct molecular weight is most important. In choosing the correct band for ADAM8 and ADAM17, we followed the suggestions of the manufacturer of the antibodies, where ADAM17 is shown at ~75-80 kDa (see Proteintech product documentation (No 24620-1-AP)) and ADAM8 is shown at ~60-65 kDa (see Proteintech product documentation No 23778-1-AP). However, when comparing our western blots to those shown in the literature and the articles cited by the Reviewer, we agree that ADAM17 is frequently evaluated at ~95-130 kDa. Hence, we verified the correct molecular weight of ADAM17 and ADAM8 using lysates of ADAM17 and ADAM8 overexpressing HEK293 cells. ADAM17 bands were detected at ~125 kDa and 80 kDa in the overexpression lysate. However, in the heart tissue lysates used for Fig. 1a ADAM17 is detected only at 80 kDa. ADAM8 is detected at 62-75 kDa in the heart tissue lysates as well as the overexpression lysate (see Reviewer Fig. 1a below). While conducting these experiments and collecting the original blots for the Source Data file, we realized that the molecular weights of ADAM12, 17 and 19 were incorrectly reported in Figure 1a of the original manuscript. This has been corrected (see below).

We agree with the Reviewer that lysis conditions are critical for assessing ADAM10 and ADAM17 expression by western blot since they undergo autodegradation. In our lysis buffer we used the cComplete protease inhibitor cocktail (product No 04 693 124 001, Roche), which according to the product information from March 2021 is suitable for the inhibition of a broad spectrum of serine, cysteine, and metalloproteases, as well as calpains. This cComplete protease inhibitor cocktail is indeed different from the cComplete inhibitor cocktail used in the publications suggested by the Reviewer, which does not contain MMP inhibitors. Nevertheless, we performed control experiments blotting ADAM10 and ADAM17 in heart tissue using our standard lysis buffer we used in the experiments as well as standard lysis buffer in combination with 1,10-phenanthroline (10mM), BB-2516 (20µM), GI254023X (5µM) as suggested in McIlwain et al., Science. 2012 Jan 13;335(6065):229-32 and Brummer, T et al. FASEB J. 2018 Jul;32(7):3560-3573. The results are presented below and show that the different lysis buffers yield similar results (see Reviewer Fig. 1b below) indicating that our standard lysis buffer containing cComplete protease inhibitor cocktail with MMP inhibitors is sufficient to block the autodegradation of

mature ADAM10 and ADAM17. We have now stated the precise cOmplete protease inhibitor cocktail we used (including product number) and that it contains also metalloprotease inhibitors in the Methods section of the revised manuscript:

Cells were homogenized in RIPA buffer containing 150 mM NaCl, 50 mM Tris-HCl, 1 mM EDTA, 1% IGEPAL CA-630, 0.25% Na-deoxycholate, 0.1% SDS (all Sigma-Aldrich), cOmplete protease inhibitor cocktail including metalloprotease inhibitors (product No 04 693 124 001, Roche) and PhosSTOP phosphatase inhibitor cocktail (Roche) by mechanical disruption using the TissueLyzer LT bead mill (Qiagen) for 4 min at a frequency of 50/s.

Reviewer Fig. 1. a Western blot analysis for ADAM8 and ADAM17. Correct molecular weight was determined using specific overexpression lysates. **b** Western blot analysis of 4 independent heart tissue lysates. Proteins were isolated using the indicated lysis condition.

Fig. 1. ADAM10 is upregulated in experimental heart injury as well as patients with ischemic cardiomyopathy and correlates with ANP/BNP expression. **a** Western blot analysis and **b** quantification of indicated ADAMs in heart tissue lysates of sham-operated (Sham) and LAD-ligated (MI) mice 3 days after infarction ($n = 5$, mean \pm SEM, ns=not significant, $***P < 0.001$, two-tailed t test). **c** Western blot analysis of ADAM10 in heart tissue lysates of from patients with ischemic cardiomyopathy (ICM) and dilated cardiomyopathy (DCM) versus non-failing controls (NF) ($n = 6$). **d-e** Pearson's correlation of ADAM10 mRNA expression against expression of NPPA (ANP) and **e** NPPB (BNP) ($n = 12$). **f** Western blot analysis and **g** quantification of ADAM10 in heart tissue lysates of mice 14 days after sham-operation (Sham) and LAD-ligation (MI) ($n = 6$, mean \pm SEM, $**P < 0.01$, two-tailed t test). **h** Immunofluorescence images and quantification of ADAM10 (A10), cardiac Troponin T (cTnT, cardiomyocytes, CM), Vimentin (Vim, fibroblasts, FB), CD31 (endothelial cells, EC) and CD45 (immune cells, IC) stained heart tissue sections of sham-operated (Sham) and LAD-ligated (MI) mice 14 days after surgery. Nuclei are stained with DAPI. Representative images are shown. Scale bar – upper panel, 20 μ m. Scale bar – lower panel, 10 μ m. Dotted line indicates the infarct area. $n = 4$ per group, mean \pm SEM, ns = not significant, $*P < 0.05$, Mann-Whitney test with Dunn's posttest. **i** Gene expression signature of ADAM10 (A10) in indicated cell types of human hearts. Publicly available single cell sequencing data of the "Heart Cell Atlas" were analyzed.

2) The authors should provide evidence that 100mg/kg of the A10i is indeed a selective concentration for ADAM10 over ADAM17, for example by testing how this concentration affects the release of TNF α from LPS-stimulated BMDM, which is mainly dependent on ADAM17. In general, the authors should discuss the effects of the A10i instead of the “effect of ADAM10 inhibition”, since this is only a partially selective inhibitor that can also block other ADAMs and MMPs.

We agree with the Reviewer that for the experiments using the ADAM10 inhibitor, the specificity for ADAM10 over ADAM17 should be assured. We decided to address this point in two ways:

(1) we markedly extended our experiments in the ADAM10 KO mice, including a completely new series of experiments. The series addresses the effect of the ADAM10 KO three days after MI. The level of IL-1 β in the infarcted area is lower in ADAM10 KO and the expression of non-shed CX3CL1 is higher (**Fig 6e, f of the revised manuscript, see below**). Using FACS, we show that like in the case for using the ADAM10 inhibitor, the infiltration of the infarcted myocardium by neutrophils is significantly lower in ADAM10 KO than in WT, as well as the neutrophil number in the blood (**Fig. 6g, h of the revised manuscript, see below**). Moreover, already at three days, pump function is better compared to WT, as was seen in the animals treated with the ADAM10 inhibitor (**Fig. 6b of the revised manuscript, see below**), which is retained at day 14 (**Fig. 6c of the revised manuscript, see below**). These experiments indicate that the effects observed upon application of ADAM10 inhibitor are indeed mediated by ADAM10. The corresponding section in the Results reads as follows:

We next investigated the effects of the ADAM10 KO in MI. As soon as 3 days after LAD ligation, ADAM10 KO mice showed a significantly higher EF compared to WT controls. By day 14, FAS and EF as well as LVID were improved as compared to WT controls (Fig. 6b-d) validating the demonstrated cardioprotective effects of A10i. Importantly, and similar to the results of the A10i, ADAM10 KO resulted in significantly reduced shedding of CX3CL1 in the infarct and border zone and reduced cardiac expression of IL-1 β 3 days after infarction (Fig. 6e, f). ADAM10 expression was significantly reduced. Also, cardiac neutrophil invasion was significantly lower in ADAM10 KO 3 days after MI as were blood neutrophil counts (Fig. 6 g, h and Supplementary Fig. 7).

(2) Moreover, we analyzed the effect on ADAM10 inhibitor on myocardial TNF α and other published ADAM10 substrates in heart tissue lysates three days after MI. We observed only a very small reduction in TNF α levels, which was not significant. This information is now detailed in the new **Supplementary Fig. 3b (see below)** and described in the new manuscript as follows:

Other substrates of ADAM10 and also ADAM17 were not affected by the ADAM10 inhibitor (Supplementary Fig. 3b).

We agree with the Reviewer that the term “ADAM10 inhibitor” is more correct than “ADAM10 inhibition” and have changed this in the revised manuscript.

Fig. 6. Cardiomyocyte-specific ADAM10 KO improves post-MI cardiac function. **a-c** Echocardiography assessment of fractional area shortening, ejection fraction and left ventricular end-systolic interior diameter in 8-week-old ADAM10^{fl/fl} (WT) and (αMHC-Cre) ADAM10 KO mice under **a** basal conditions (WT, n = 7, KO, n = 8, mean ± SEM, ns = not significant, *P<0.05, two-tailed t test) as well as **b** 3 (WT, n = 8, KO, n = 9, mean ± SEM, ns = not significant, *P<0.05, two-tailed t test) and **c** 14 days after myocardial infarction (WT, n = 6, KO, n = 9, mean ± SEM, ns = not significant, *P<0.05, two-tailed t test). **d** Representative end-systolic B-mode and M-mode echocardiograms of ADAM10 WT and ADAM10 KO mice 14 days after myocardial infarction. **e** Western blot analysis and **f** quantification of ADAM10, CX3CL1 and IL-1 β expression in heart tissue lysates of ADAM10 WT and ADAM10 KO mice 3 days after myocardial infarction (n = 6, mean ± SEM, ns = not significant, *P<0.05, **P<0.05, two-tailed t test). **g** Analysis of leukocyte counts in blood samples of ADAM10 WT (n = 8) and ADAM10 KO (n = 7) mice (mean ± SEM, ns = not significant, *P<0.01, Mann-Whitney test). **h** Analysis of leukocyte counts in the infarcted area/ infarct border zone of ADAM10 WT (n = 8) and ADAM10 KO (n = 7) mice (mean ± SEM, ns = not significant, *P<0.01, Mann-Whitney test).

Supplementary Fig. 7. Cardiomyocyte-specific ADAM10 knockout reduces neutrophil bone marrow egress and heart tissue infiltration following myocardial infarction. Gating strategy for identification of neutrophils (CD45⁺, CD11b⁺, Ly6G⁺), eosinophils (CD45⁺, CD11b⁺, Ly6G⁺, SigF⁺), monocytes (CD45⁺, CD11b⁺, Ly6G⁺, SigF⁺, F4/80⁺, Ly6C⁺) and macrophages (CD45⁺, CD11b⁺, Ly6G⁺, SigF⁺, F4/80⁺) in the infarcted area/ infarct border zone of ADAM10 KO (n = 7) and WT (n = 8) treated mice 3 d after infarction.

Supplementary Fig. 3. Pharmacological ADAM10 inhibitor reduces chemotaxis-associated gene expression and CX3CL1 shedding. **a** Quantitative real-time PCR of significantly downregulated neutrophil chemotaxis-associated genes in heart tissue of A10i (n= 5) and DMSO (n = 4) treated mice 3 d after MI (mean ± SEM, *P<0.05, Mann-Whitney test with Dunn's posttest). **b** Western blot analysis and quantification of indicated proteins in heart tissue lysates of GI254023X (A10i) and DMSO treated mice 3 d after myocardial infarction (n = 3, mean ± SEM, ns = not significant, *P<0.05, Mann-Whitney test with Dunn's posttest).

3) The immunofluorescence analysis in Figure 2 does not provide rigorous evidence for the activation state of ADAM10, and the cited JBC paper uses mainly a chimera between the ADAM10 cytoplasmic domain and Tac. It is also not clear how a hydroxamate type inhibitor such as the A10i should affect the maturation or subcellular localization of ADAM10? Figure 2h and the corresponding results section should be removed, as it is not clear that these experiments support the authors interpretation.

We agree with the Reviewer and have removed Figure 2h and the corresponding Results section from the revised manuscript.

4) In this reviewer's opinion, the discussion is too focused on ADAM10/CX3CL1 signaling as the main target of the A10i and a main pathogenic mechanism in MI. It is also possible that the effect of the A10i on other processes, such as shedding of other substrates of ADAM10 or ADAM17 or inhibition of other ADAMs or MMPs, offer the described protection from MI. The blockade of IL-1 β is a case in point.

We agree with the Reviewer that also other mechanisms in addition to CX3CL1 could contribute to the effect of the ADAM10 inhibitor and ADAM10 KO. We have added **Supplemental Fig. 3b (see above)**, depicting that in heart tissue lysates neither FasL, Notch1, N-cadherin or CXCL16 are affected by the ADAM10 inhibitor three days after MI. Regarding the reduced IL-1 β levels, we regard this as a downstream event caused by the reduced neutrophil invasion, since IL-1 β has not been described as a direct target of ADAM10 (or ADAM17). We further assure the critical relevance of ADAM10 by adding an additional series of experiments in the ADAM10 KO animals three and 14 days after MI, which confirm our results obtained with the ADAM10 inhibitor (**see response to point 2 above**). We have included a section to the Discussion addressing the contribution of other ADAMs and signaling molecules reading as follows:

In addition to CX3CL1 also other processes may contribute to the effects of A10i and ADAM10 KO. We therefore probed for a number of well-known ADAM10 (or ADAM17) targets which were not affected by pharmacological ADAM10 inhibition after MI. This does not exclude the involvement of other pathways not explored in our experiments. Moreover, single cell transcriptomics would allow a more comprehensive description of the actions of specific cell types in the processes occurring in the infarct and border zone.

5) The cardiomyocytes used to determine the expression of ADAM10 were derived from IPSC, so it would be good to independently verify the statement that "cardiac ADAM10 is mostly expressed in cardiomyocytes" (top of page 13) using cardiomyocytes isolated from mouse hearts, or other approaches, such as immunofluorescence analysis or expression analysis (scSEQ or qPCR from FACS-sorted or bead isolated heart cell types) to support this conclusion.

We agree with the Reviewer that it is important to clearly define the expression of ADAM10 in heart tissue. Therefore, we have performed immunostaining in cardiac slices of control and MI hearts 14 days after MI. Probing for ADAM10, CX3CL1, cTnT (cardiomyocytes), Vimentin (fibroblasts), CD31 (endothelial cells) and CD45 (leukocytes) we found that ADAM10 is predominantly expressed in cardiomyocytes and endothelial cells in the heart in Sham as well as 14 days after MI (**Fig. 1h of the revised manuscript, see below**). CX3CL1 is predominantly expressed in cardiomyocytes and endothelial cells (**Fig. 5a, b and Supplementary Fig. 5a, b of the revised manuscript, see below**). Quantitative analysis revealed that in sham-operated mice 66.38 \pm 9.74% of cTnT-positive cardiomyocytes, 21.88 \pm 3.23% of Vimentin-positive fibroblasts, 99.35 \pm 0.84% of CD31-positive endothelial cells and 8.98 \pm 3.42% of CD45-positive immune cells were positive for ADAM10. Moreover, we detected an increase in the number of ADAM10-positive cardiomyocytes when we examined heart tissue section of LAD-ligated mice 14 d after infarction. In addition, we verified cardiac ADAM10 expression by analysis of publicly available single cell sequencing data from human hearts (**Fig. 1i of the revised manuscript, see below**). The corresponding section in the results section reads as follows: **Histological analysis of sham-operated and LAD-ligated mice 14 days after infarction and screening of a panel of human induced pluripotent stem cell-derived cardiomyocytes (iCM), mouse immortalized cardiomyocytes (HL-1), human atrial and ventricular fibroblasts (HAF and HVF) and human endothelial cells (EC) revealed that cardiomyocytes as well as endothelial cells exhibit the highest ADAM10 expression (Fig. 1h and Supplementary Fig. 1b-g). This observation was confirmed by analysis of publicly available human single-cell sequencing gene expression data of the "Heart Cell Atlas" ²³ (Fig. 1i and Supplementary Fig. 1d). However, cardiomyocytes were the only cell type where ADAM10 expression was induced upon infarction (Fig. 1h). Mimicking hypoxia with the HIF hydroxylase inhibitor dimethylxaloylglycine (DMOG) led to induction of mature ADAM10 in all tested cardiomyocyte-**

samples, which was less pronounced in endothelial cells and absent in cardiac fibroblasts (Supplementary Fig. 1h, i).

Fig. 1. ADAM10 is upregulated in experimental heart injury as well as patients with ischemic cardiomyopathy and correlates with ANP/BNP expression. **a** Western blot analysis and **b** quantification of indicated ADAMs in heart tissue lysates of sham-operated (Sham) and LAD-ligated (MI) mice 3 days after infarction ($n = 5$, mean \pm SEM, ns=not significant, *** $P < 0.001$, two-tailed t test). **c** Western blot analysis of ADAM10 in heart tissue lysates of from patients with ischemic cardiomyopathy (ICM) and dilated cardiomyopathy (DCM) versus non-failing controls (NF) ($n = 6$). **d-e** Pearson's correlation of ADAM10 mRNA expression against expression of NPPA (ANP) and **e** NPPB (BNP) ($n = 12$). **f** Western blot analysis and **g** quantification of ADAM10 in heart tissue lysates of mice 14 days after sham-operation (Sham) and LAD-ligation (MI) ($n = 6$, mean \pm SEM, ** $P < 0.01$, two-tailed t test). **h** Immunofluorescence images and quantification of ADAM10 (A10), cardiac Troponin T (cTnT, cardiomyocytes, CM), Vimentin (Vim, fibroblasts, FB), CD31 (endothelial cells, EC) and CD45 (immune cells, IC) stained heart tissue sections of sham-operated (Sham) and LAD-ligated (MI) mice 14 days after surgery. Nuclei are stained with DAPI. Representative images are shown. Scale bar – upper panel, 20 μ m. Scale bar – lower panel, 10 μ m. Dotted line indicates the infarct area. $n = 4$ per group, mean \pm SEM, ns = not significant, * $P < 0.05$, Mann-Whitney test with Dunn's posttest. **i** Gene expression signature of ADAM10 (A10) in indicated cell types of human hearts. Publicly available single cell sequencing data of the "Heart Cell Atlas" were analyzed.

6) Regarding Figure 5 b-d, if CX3CL1 is shed by ADAM10, shouldn't there be more in the presence of the A10i inhibitor? Moreover, it is not clear how and why an A10 inhibitor would block the co-IP of the enzyme and substrate? And finally, the IF data in Figure 5d are also difficult to interpret.

After deeply considering this comment, we have to agree with the Reviewer that the experiments depicted in Figure 5 b-d are really not easy to interpret. Therefore, we undertook steps to experimentally clarify our results as well as to improve the presentation. With regard to Figure 5b, when looking at full-length CX3CL1 at ~85 kDa using an N-terminal antibody, we indeed see the result the Reviewer expects: Upon application of the ADAM10 inhibitor there is more cellular CX3CL1 since less is shed. We concur that it is preferable to show the impact on ADAM10 inhibition on the full-length protein and have changed **Fig. 5d and e in the revised manuscript (see below)** accordingly. This result fits well to **Fig. 5f of the revised manuscript (see below)**, where we show that there is less soluble CX3CL1 in the cell culture medium upon ADAM10 inhibition. We have removed the immunofluorescence data formerly depicted in Figure 5d, since it indeed provided little insight. Regarding the co-immunoprecipitation in Figure 5e and f, we found the indication that the ADAM10 inhibitor reduces the interaction/binding between ADAM10 and its substrate interesting and worth reporting. We however agree that to make this claim convincing, further experiments are needed. Since this is not in the focus of the current manuscript, we decided to redo the immunoprecipitation without the ADAM10 inhibitor experiment to only show the interaction between ADAM10 and CX3CL1 (**Fig. 5c of the revised manuscript, see below**). We hope the Reviewer can agree to this course of action.

We have moreover used this new approach for assessing CX3CL1 shedding after MI (**Fig. 3i, j of the revised manuscript, see below**). Our data indicate that CX3CL1 shedding is increased three days, but not 14 or 28 days, after MI which is prevented by the ADAM10 inhibitor. Moreover, in cardiomyocyte-specific ADAM10 KO, our data indicate lower CX3CL1 shedding three days after MI than in WT (**Fig. 6e, f of the revised manuscript, see below**).

This led to the following changes in the Results of the revised manuscript:

We next assessed cardiac expression of CX3CL1 and IL-1 β at 3, 14 and 28 days post myocardial infarction using immunoblot analysis. Evaluating full-length (non-shed) CX3CL1 using an N-terminal antibody, we found CX3CL1 to be decreased at 3 days post MI, consistent with increased shedding of ADAM10, that was no longer detectable on day 14 and day 28 (Fig.3i, j). In contrast, IL-1 β remained elevated through day 28 (Fig. 3i, k). Both changes were sensitive to treatment with A10i (Fig 3i-k).

Importantly, and similar to the results of the A10i, ADAM10 KO resulted in significantly reduced shedding of CX3CL1 in the infarct and border zone and reduced cardiac expression of IL-1 β 3 days after infarction (Fig. 6e, f). ADAM10 expression was significantly reduced.

Fig. 5. Cardiomyocyte-specific ADAM10/CX3CL1 signaling regulates neutrophil migration. **a** Immunofluorescence images and **b** quantification of CX3CL1, cardiac Troponin T (cTnT - cardiomyocytes) and CD31 (endothelial cells) stained heart tissue sections of sham-operated (Sham) and LAD-ligated (MI) mice 14 days after surgery. Nuclei are stained with DAPI. Representative images are shown. Scale bar, 20 μ m. $n = 4$ per group, mean \pm SEM, ns = not significant, Mann-Whitney test with Dunn's posttest. **c** Co-immunoprecipitation of endogenous ADAM10 and CX3CL1 from whole-cell lysates of HL-1 cells. Representative western blots are shown ($n = 4$). IgG, immunoglobulin G as a control. WCL, whole cell lysate. **d** Western blot analysis of CX3CL1 and **e** quantification of CX3CL1 expression ($n = 6$ per group, mean \pm SEM, ns = not significant, $**P < 0.01$, $***P < 0.001$, one-way ANOVA with Tukey's posttest) as well as **f** ectodomain shedding (CX3CL1 levels in supernatant) in normoxic as well as GI254023X (A10i) and DMSO treated hypoxic (1% O_2) mouse cardiomyocytes (HL-1) ($n = 3$ per group, mean \pm SEM, ns = not significant, $**P < 0.01$, Kruskal-Wallis test with Dunn's posttest). **g** Treatment scheme. HL-1 cells were cultured for 3 h under normoxic or hypoxic conditions in parallel to A10i or DMSO treatment in combination with unspecific control siRNA (siC) or CX3CL1 depletion (siCX3CL1). Supernatants were used in the lower chamber and 3×10^5 freshly isolated mouse bone marrow derived neutrophils or macrophages were seeded in the upper chamber of transwell plates (8 μ m pore size) to determine migration capacity. **h** Western blot analysis and **i** quantification of CX3CL1 expression in HL-1 cells treated for 48 h with unspecific control siRNA (siC) or CX3CL1-specific siRNA (siCX3#1 and siCX3#2). Representative western blots ($n = 3$ per group, mean \pm SEM, $**P < 0.01$, Kruskal-Wallis test with Dunn's posttest) are shown. **j-k** Transwell migration assays of bone marrow derived **j** neutrophils and **k** macrophages using the supernatants of HL-1 cells that were cultured under hypoxic conditions in parallel to A10i or DMSO treatment in combination with unspecific control siRNA (siC) or CX3CL1 depletion (siCX3CL1#1 and CX3CL1#2) as chemoattractant ($n = 3$ per group, mean \pm SEM, ns = not significant, $**P < 0.01$, Kruskal-Wallis test with Dunn's posttest).

Fig. 3. ADAM10 inhibition after MI reduces neutrophil chemotaxis-associated gene expression and CX3CL1 serum levels. **a** Principal components analysis (PCA) of transcriptomes of sham-operated (Sham, $n = 3$), GI254023X (A10i, $n = 4$) and DMSO ($n = 4$) treated mice 3 days after myocardial infarction. Principal components were computed for each sample using gene expression. The first two principal components (PC1 and PC2) are shown. The percent of variance explained by each component is reported in parentheses. **b** Volcano plot representing the expression changes of all genes. Significantly down- and upregulated genes ($FDR < 0.01$) are colored navy and magenta, respectively. Genes that do not show significant expression changes are colored gray. Examples of neutrophil chemotaxis-associated genes are labeled with gene names. **c-d** Functional annotation clustering for the gene ontology (GO) terms **c** biological process and **d** molecular function of genes with significantly decreased expression upon A10i treatment. **e-f** Gene set enrichment analysis of genes with significantly decreased expression upon A10i treatment performed on GO terms. **g** In-depth analysis of the significantly downregulated genes of the indicated functional annotation clusters. **h** Quantification of CXCL1 ($n = 4$ vs. 5), IL-6 ($n = 6$ vs. 5), CXCL16 ($n = 10$ vs. 8), CX3CL1 ($n = 9$ vs. 9), IL-1 β ($n = 8$ vs. 8) and CXCL5 ($n = 8$ vs. 8) serum levels in A10i ($n = 5$) and DMSO ($n = 4$) treated mice 3 days after MI (mean \pm SEM, ns = not significant, ** $P < 0.01$, *** $P < 0.001$, Mann-Whitney test). **i** Western blot analysis and **j-k** quantification of **j** CX3CL1 and **k** IL-1 β in heart tissue lysates of sham-operated (Sham) and LAD-ligated (MI) mice 3 days after infarction ($n = 5$, mean \pm SEM, ns = not significant, * $P < 0.05$, ** $P < 0.01$, Kruskal-Wallis test with Dunn's posttest). **l** Western blot analysis and **m** quantification of CX3CL1 in heart tissue lysates from patients with ischemic cardiomyopathy (ICM) and non-failing controls (NF) ($n = 6$, mean \pm SEM, * $P < 0.05$, ** $P < 0.01$, two-tailed t test).

Fig. 6. Cardiomyocyte-specific ADAM10 KO improves post-MI cardiac function. **a-c** Echocardiography assessment of fractional area shortening, ejection fraction and left ventricular end-systolic interior diameter in 8-week-old ADAM10^{fl/fl} (WT) and (αMHC-Cre) ADAM10 KO mice under **a** basal conditions (WT, n = 7, KO, n = 8, mean ± SEM, ns = not significant, *P<0.05, two-tailed t test) as well as **b** 3 (WT, n = 8, KO, n = 9, mean ± SEM, ns = not significant, *P<0.05, two-tailed t test) and **c** 14 days after myocardial infarction (WT, n = 6, KO, n = 9, mean ± SEM, ns = not significant, *P<0.05, two-tailed t test). **d** Representative end-systolic B-mode and M-mode echocardiograms of ADAM10 WT and ADAM10 KO mice 14 days after myocardial infarction. **e** Western blot analysis and **f** quantification of ADAM10, CX3CL1 and IL-1β expression in heart tissue lysates of ADAM10 WT and ADAM10 KO mice 3 days after myocardial infarction (n = 6, mean ± SEM, ns = not significant, *P<0.05, **P<0.05, two-tailed t test). **g** Analysis of leukocyte counts in blood samples of ADAM10 WT (n = 8) and ADAM10 KO (n = 7) mice (mean ± SEM, ns = not significant, *P<0.01, Mann-Whitney test). **h** Analysis of leukocyte counts in the infarcted area/ infarct border zone of ADAM10 WT (n = 8) and ADAM10 KO (n = 7) mice (mean ± SEM, ns = not significant, *P<0.01, Mann-Whitney test).

7) Regarding the Western blots in Supplementary Figure 5A, there is large variability in the p-PLB samples at 6 m. How meaningful are these data? The actual blots for the MI samples at 14d are not much more convincing, with such large variability between samples that it is difficult to draw conclusions about the biological significance of these findings

We agree with the Reviewer and have removed these figures and the corresponding passages in the Results section from the revised manuscript.

8) Many key findings were not addressed in the conditional cardiomyocyte ADAM10 ko mice, such as CX3CL1 shedding and neutrophil recruitment.

According to the suggestions of the Reviewer we have added a completely new series of experiments performed on the cardiomyocyte-specific ADAM10 KO mice to the revised manuscript. It addresses the effect of the ADAM10 KO three days after MI. The level of IL-1 β in the infarcted area is lower in ADAM10 KO and the expression of non-shed CX3CL1 is higher (**Fig. 6e, f of the revised manuscript, see above**). Using FACS, we show that like in the case for using the ADAM10 inhibitor, the infiltration of the infarcted myocardium by neutrophils is significantly lower in ADAM10 KO than in WT, as well as the neutrophil number in the blood (**Fig. 6g, h of the revised manuscript, see above**). Moreover, already at three days, pump function is better compared to WT, as was seen in the animals treated with the ADAM10 inhibitor (**Fig. 6b of the revised manuscript, see above**), which is retained at day 14 (**Fig. 6c of the revised manuscript, see above**). The corresponding section in the Results reads as follows:

We next investigated the effects of the ADAM10 KO in MI. As soon as 3 days after LAD ligation, ADAM10 KO mice showed a significantly higher EF compared to WT controls. By day 14, FAS and EF as well as LVID were improved as compared to WT controls (Fig. 6b-d) validating the demonstrated cardioprotective effects of A10i. Importantly, and similar to the results of the A10i, ADAM10 KO resulted in significantly reduced shedding of CX3CL1 in the infarct and border zone and reduced cardiac expression of IL-1 β 3 days after infarction (Fig. 6e, f). ADAM10 expression was significantly reduced. Also, cardiac neutrophil invasion was significantly lower in ADAM10 KO 3 days after MI as were blood neutrophil counts (Fig. 6 g, h and Supplementary Fig. 7).

9) *Re the model shown in Supplementary Figure 6, there is no meaningful discussion of the transcriptional regulation of the various genes that are affected by the A10i. Only CX3CL1 shedding is mentioned, although there are many other possible mechanisms for how the A10i could affect the outcome of the LAD MI model.*

We agree with the Reviewer that the model depicted in Supplementary Figure 6 is over-simplified. Consequently, we have removed this figure from the revised manuscript and instead added a section to the Discussion addressing the possible mechanisms of how the ADAM10 inhibitor could affect the outcome reading:

In addition to CX3CL1 also other processes may contribute to the effects of A10i and ADAM10 KO. We therefore probed for a number of well-known ADAM10 (or ADAM17) targets which were not affected by pharmacological ADAM10 inhibition after MI. This does not exclude the involvement of other pathways not explored in our experiments. Moreover, single cell transcriptomics would allow a more comprehensive description of the actions of specific cell types in the processes occurring in the infarct and border zone.

Reviewer #3 (Remarks to the Author):

This study provides evidence that pharmacological inhibition and myocyte-specific deletion of ADAM10 improves cardiac remodeling after myocardial infarction. This goes along with diminished CX3CL1 ectodomain shedding and reduced neutrophil infiltration into the injured heart. There are, however, some conceptual and experimental shortcomings which finally question the magnitude and biological relevance of the observed effect and the cellular subpopulation involved in the infarcted heart.

We thank the Reviewer for critically reviewing our manuscript and the helpful comments and suggestions.

Major

a. *The assignment and cellular distribution of ADAM10 is not convincing. The authors have used iPS-derived CM and a mouse cell line resembling features of CM but not CM freshly isolated from control and infarcted hearts. Similarly, the fibroblasts used are likely not to resemble the activated fibroblasts present in the infarcted heart. Therefore, it is quite possible that aside of ADAM10 in cardiomyocytes other non-cardiomyocytes contribute to the ADAM10 signal in the infarcted heart. What is needed is a direct comparison of freshly isolated mouse CM with FACS- sorted non-CM (endothelium, fibroblasts, immune cells). Helpful in this regard would also be a search on the cellular distribution of expressed ADAM10 in published single cell sequencing data sets. This in part may resolve the question of cellular specificity.*

We agree with the Reviewer that it is important to clearly define the expression of ADAM10 in heart tissue. Therefore, we have performed immunostaining in cardiac slices of control and MI hearts 14 days after MI. Probing for ADAM10, CX3CL1, cTnT (cardiomyocytes), Vimentin (fibroblasts), CD31 (endothelial cells) and CD45 (leukocytes) we found that ADAM10 is predominantly expressed in cardiomyocytes and endothelial cells in the heart in Sham as well as 14 days after MI (**Fig. 1h of the revised manuscript, see below**). CX3CL1 is predominantly expressed in cardiomyocytes and endothelial cells (**Fig. 5a, b and Supplementary Fig. 5a, b of the revised manuscript, see below**). Quantitative analysis revealed that in sham-operated mice 66.38±9.74% of cTnT-positive cardiomyocytes, 21.88±3.23% of Vimentin-positive fibroblasts, 99.35±0.84% of CD31-positive endothelial cells and 8.98±3.42% of CD45-positive immune cells were positive for ADAM10. Moreover, we detected an increase in the number of ADAM10-positive cardiomyocytes when we examined heart tissue section of LAD-ligated mice 14 d after infarction. In addition, we verified cardiac ADAM10 expression by analysis of publicly available human single cell sequencing data (**Fig. 1i of the revised manuscript, see below**). The corresponding section in the results section reads as follows:

Histological analysis of sham-operated and LAD-ligated mice 14 days after infarction and screening of a panel of human induced pluripotent stem cell-derived cardiomyocytes (iCM), mouse immortalized cardiomyocytes (HL-1), human atrial and ventricular fibroblasts (HAF and HVF) and human endothelial cells (EC) revealed that cardiomyocytes as well as endothelial cells exhibit the highest ADAM10 expression (Fig. 1h and Supplementary Fig. 1b-g). This observation was confirmed by analysis of publicly available human single-cell sequencing gene expression data of the "Heart Cell Atlas"²³ (Fig. 1i and Supplementary Fig. 1d). However, cardiomyocytes were the only cell type where ADAM10 expression was induced upon infarction (Fig. 1h). Mimicking hypoxia with the HIF hydroxylase inhibitor dimethylxaloylglycine (DMOG) led to induction of mature ADAM10 in all tested cardiomyocyte-samples, which was less pronounced in endothelial cells and absent in cardiac fibroblasts (Supplementary Fig. 1h, i).

Fig. 1. ADAM10 is upregulated in experimental heart injury as well as patients with ischemic cardiomyopathy and correlates with ANP/BNP expression. **a** Western blot analysis and **b** quantification of indicated ADAMs in heart tissue lysates of sham-operated (Sham) and LAD-ligated (MI) mice 3 days after infarction ($n = 5$, mean \pm SEM, ns=not significant, $***P < 0.001$, two-tailed t test). **c** Western blot analysis of ADAM10 in heart tissue lysates of from patients with ischemic cardiomyopathy (ICM) and dilated cardiomyopathy (DCM) versus non-failing controls (NF) ($n = 6$). **d-e** Pearson's correlation of ADAM10 mRNA expression against expression of NPPA (ANP) and **e** NPPB (BNP) ($n = 12$). **f** Western blot analysis and **g** quantification of ADAM10 in heart tissue lysates of mice 14 days after sham-operation (Sham) and LAD-ligation (MI) ($n = 6$, mean \pm SEM, $**P < 0.01$, two-tailed t test). **h** Immunofluorescence images and quantification of ADAM10 (A10), cardiac Troponin T (cTnT, cardiomyocytes, CM), Vimentin (Vim, fibroblasts, FB), CD31 (endothelial cells, EC) and CD45 (immune cells, IC) stained heart tissue sections of sham-operated (Sham) and LAD-ligated (MI) mice 14 days after surgery. Nuclei are stained with DAPI. Representative images are shown. Scale bar – upper panel, 20 μ m. Scale bar – lower panel, 10 μ m. Dotted line indicates the infarct area. $n = 4$ per group, mean \pm SEM, ns = not significant, $*P < 0.05$, Mann-Whitney test with Dunn's posttest. **i** Gene expression signature of ADAM10 (A10) in indicated cell types of human hearts. Publicly available single cell sequencing data of the "Heart Cell Atlas" were analyzed.

Fig. 5. Cardiomyocyte-specific ADAM10/CX3CL1 signaling regulates neutrophil migration. **a** Immunofluorescence images and **b** quantification of CX3CL1, cardiac Troponin T (cTnT - cardiomyocytes) and CD31 (endothelial cells) stained heart tissue sections of sham-operated (Sham) and LAD-ligated (MI) mice 14 days after surgery. Nuclei are stained with DAPI. Representative images are shown. Scale bar, 20 μ m. n = 4 per group, mean \pm SEM, ns = not significant, Mann-Whitney test with Dunn's posttest. **c** Co-immunoprecipitation of endogenous ADAM10 and CX3CL1 from whole-cell lysates of HL-1 cells. Representative western blots are shown (n = 4). IgG, immunoglobulin G as a control. WCL, whole cell lysate. **d** Western blot analysis of CX3CL1 and **e** quantification of CX3CL1 expression (n = 6 per group, mean \pm SEM, ns = not significant, **P < 0.01, ***P < 0.001, one-way ANOVA with Tukey's posttest) as well as **f** ectodomain shedding (CX3CL1 levels in supernatant) in normoxic as well as A10i and DMSO treated hypoxic (1% O₂) mouse cardiomyocytes (HL-1) (n = 3 per group, mean \pm SEM, ns = not significant, **P < 0.01, Kruskal-Wallis test with Dunn's posttest). **g** Treatment scheme. HL-1 cells were cultured for 3 h under normoxic or hypoxic conditions in parallel to A10i or DMSO treatment in combination with unspecific control siRNA (siC) or CX3CL1 depletion (siCX3CL1). Supernatants were used in the lower chamber and 3x10⁵ freshly isolated mouse bone marrow derived neutrophils or macrophages were seeded in the upper chamber of transwell plates (8 μ m pore size) to determine migration capacity. **h** Western blot analysis and **i** quantification of CX3CL1 expression in HL-1 cells treated for 48 h with unspecific control siRNA (siC) or CX3CL1-specific siRNA (siCX3#1 and siCX3#2). Representative western blots (n = 3 per group, mean \pm SEM, **P < 0.01, Kruskal-Wallis test with Dunn's posttest) are shown. **j-k** Transwell migration assays of bone marrow derived **j** neutrophils and **k** macrophages using the supernatants of HL-1 cells that were cultured under hypoxic conditions in parallel to A10i or DMSO treatment in combination with unspecific control siRNA (siC) or CX3CL1 depletion (siCX3CL1#1 and CX3CL1#2) as chemoattractant (n = 3 per group, mean \pm SEM, ns = not significant, **P < 0.01, Kruskal-Wallis test with Dunn's posttest).

Supplementary Fig. 5. CX3CL1 is expressed in cardiomyocytes and regulates neutrophil transmigration upon hypoxia. **a** Immunofluorescence images and **b** quantification of CX3CL1, Vimentin (Vim - fibroblasts) and CD45 (immune cells) stained heart tissue sections of sham-operated (Sham) and LAD-ligated (MI) mice 14 d after surgery. Nuclei are stained with DAPI. Representative images are shown. Scale bar, 20 μ m. $n = 4$ per group, mean \pm SEM, ns = not significant, Mann-Whitney test with Dunn's posttest. **c** Western blot analysis of CX3CL1 expression in human iPSC-derived cardiomyocytes (iCM1), human ventricular fibroblasts (HVF), human umbilical vein endothelial cells (EC1) and human monocytic cells (THP-1). Representative western blots ($n = 3$ per group) are shown. CM, cardiomyocytes. FB, fibroblasts. EC, endothelial cells. MC, monocytes. **d** Quantification of CX3CL1 mRNA expression in human iPSC-derived cardiomyocytes (iCM1-2), human ventricular fibroblasts (HVF), human umbilical vein endothelial cells (EC1) and human monocytic cells (THP-1) ($n = 3$, mean \pm SEM, ns = not significant, ** $P < 0.01$, one-way ANOVA with Tukey's posttest). CM, cardiomyocytes. FB, fibroblasts. EC, endothelial cells. MC, monocytes. **e** Transwell migration assays of bone marrow derived neutrophils using the supernatants of normoxic as well as A10i and DMSO treated hypoxic HL-1 cells as chemoattractant ($n = 3$ per group, mean \pm SEM, ns = not significant, *** $P < 0.001$, Kruskal-Wallis test with Dunn's posttest).

Also note that the recruitment of blood leukocytes across the endothelium to sites of tissue inflammation was reported to involve disintegrin and ADAM10 that was implicated in leukocyte transmigration by proteolytically cleaving its endothelial substrates (DOI: 10.4049/jimmunol.1600713)

We thank the Reviewer for binging up this important aspect. Indeed, this could be another mechanism of action of the ADAM10 inhibitor we used adding to the impact of cardiomyocyte ADAM10 inhibition. To clarify this point, we have considerably extended our experiments in the ADAM10 KO mice and added two completely new series of experiments performed on the cardiomyocyte-specific ADAM10 knockout mice (ADAM10 KO) to the revised manuscript. The first series addresses the effect of three days MI in the ADAM10 KO. Using FACS, we show that like in the case for using the ADAM10 inhibitor, the infiltration of the infarcted myocardium by neutrophils is significantly lower in ADAM10 KO than in WT, as well as the neutrophil number in the blood (**Fig. 6g, h of the revised manuscript, see below**). Moreover, already at three days, pump function is better compared to WT, as was seen in the animals treated with ADAM10 inhibitor (**Fig. 6b of the revised manuscript, see below**), which is retained at day 14 (**Fig. 6c of the revised manuscript, see below**). To evaluate whether these effects could rely on reduced CX3CL1 shedding in ADAM10 KO mice, we detected N-terminal CX3CL1 in heart tissue lysates of ADAM10 KO and wildtype mice 3 days post MI. Our data indicate that the level of IL-1 β in the infarcted area is lower in ADAM10 KO and the expression of non-shed CX3CL1 is higher (**Fig. 6e, f of the revised manuscript, see below**). These new data clearly speak for cardiomyocyte ADAM10 inhibition as the predominant mechanism of action of the ADAM10 inhibitor. Importantly, we have included a new section to the Discussion of the revised manuscript addressing this point which reads as follows:

We next investigated the effects of the ADAM10 KO in MI. As soon as 3 days after LAD ligation, ADAM10 KO mice showed a significantly higher EF compared to WT controls. By day 14, FAS and EF as well as LVID were improved as compared to WT controls (Fig. 6b-d) validating the demonstrated cardioprotective effects of A10i. Importantly, and similar to the results of the A10i, ADAM10 KO resulted in significantly reduced shedding of CX3CL1 in the infarct and border zone and reduced cardiac expression of IL-1 β 3 days after infarction (Fig. 6e, f). ADAM10 expression was significantly reduced. Also, cardiac neutrophil invasion was significantly lower in ADAM10 KO 3 days after MI as were blood neutrophil counts (Fig. 6 g, h and Supplementary Fig. 7).

Additionally, we have added a section to the Discussion addressing the possible mechanisms of how the ADAM10 inhibitor could affect the outcome reading:

In addition to CX3CL1 also other processes may contribute to the effects of A10i and ADAM10 KO. We therefore probed for a number of well-known ADAM10 (or ADAM17) targets which were not affected by pharmacological ADAM10 inhibition after MI. This does not exclude the involvement of other pathways not explored in our experiments. Moreover, single cell transcriptomics would allow a more comprehensive description of the actions of specific cell types in the processes occurring in the infarct and border zone.

Fig. 6. Cardiomyocyte-specific ADAM10 KO improves post-MI cardiac function. **a-c** Echocardiography assessment of fractional area shortening, ejection fraction and left ventricular end-systolic interior diameter in 8-week-old ADAM10^{fl/fl} (WT) and (α MHC-Cre) ADAM10 KO mice under **a** basal conditions (WT, n = 7, KO, n = 8, mean \pm SEM, ns = not significant, *P<0.05, two-tailed t test) as well as **b** 3 (WT, n = 8, KO, n = 9, mean \pm SEM, ns = not significant, *P<0.05, two-tailed t test) and **c** 14 days after myocardial infarction (WT, n = 6, KO, n = 9, mean \pm SEM, ns = not significant, *P<0.05, two-tailed t test). **d** Representative end-systolic B-mode and M-mode echocardiograms of ADAM10 WT and ADAM10 KO mice 14 days after myocardial infarction. **e** Western blot analysis and **f** quantification of ADAM10, CX3CL1 and IL-1 β expression in heart tissue lysates of ADAM10 WT and ADAM10 KO mice 3 days after myocardial infarction (n = 6, mean \pm SEM, ns = not significant, *P<0.05, **P<0.05, two-tailed t test). **g** Analysis of leukocyte counts in blood samples of ADAM10 WT (n = 8) and ADAM10 KO (n = 7) mice (mean \pm SEM, ns = not significant, *P<0.01, Mann-Whitney test). **h** Analysis of leukocyte counts in the infarcted area/ infarct border zone of ADAM10 WT (n = 8) and ADAM10 KO (n = 7) mice (mean \pm SEM, ns = not significant, *P<0.01, Mann-Whitney test).

Supplementary Fig. 7. Cardiomyocyte-specific ADAM10 knockout reduces neutrophil bone marrow egress and heart tissue infiltration following myocardial infarction. Gating strategy for identification of neutrophils (CD45⁺, CD11b⁺, Ly6G⁺), eosinophils (CD45⁺, CD11b⁺, Ly6G⁻, SigF⁺), monocytes (CD45⁺, CD11b⁺, Ly6G⁻, SigF⁻, F4/80⁻, Ly6C⁺) and macrophages (CD45⁺, CD11b⁺, Ly6G⁻, SigF⁻, F4/80⁺) in the infarcted area/ infarct border zone of ADAM10 KO (n = 7) and WT (n = 8) treated mice 3 d after infarction.

b. Genetic deletion of ADAM10 is experimentally certainly superior to using systemic application of the experimental ADAM10 inhibitor GI254023X, because of the absence of possible pharmacologic side effects. In the genetic model the improvement of functional parameters (figure 6 e) was only rather small as compared with the massive improvement of ejection fraction observed after treatment with GI254023X.

We agree with the Reviewer that genetical deletion of ADAM10 is scientifically better suited to elucidate the pathophysiological mechanisms, while pharmacological ADAM10 inhibition better depicts the translational potential of targeting ADAM10 in acute MI. Consequently, we added a completely new series of experiments performed on the ADAM10 KO to the revised manuscript, which addresses the effect of three days MI in the ADAM10 KO. Using FACS, we show that like in the case for using the ADAM10 inhibitor, the infiltration of the infarcted myocardium by neutrophils is significantly lower in ADAM10 KO than in WT, as well as the neutrophil number in the blood (**Fig. 6g, h of the revised manuscript, see above**). To evaluate whether these effects could rely on reduced CX3CL1 shedding in ADAM10 KO mice, we detected N-terminal CX3CL1 in heart tissue lysates of ADAM10 KO and wildtype mice 3 days post MI. Our data indicate lower CX3CL1 shedding three days after MI than in WT (**Fig. 6f of the revised manuscript, see above**). Moreover, already at this early stage, pump function is better compared to WT (**Fig. 6b of the revised manuscript, see above**). Indeed, in these groups the effect of the genetic ADAM10 ablation on pump function is absolutely comparable to the effect we showed for the ADAM10 inhibitor. Regarding the effect of ADAM10 KO on the pump function in the 14-day MI group we have to state that the pump function achieved is very similar between ADAM10 inhibitor and ADAM10 KO (EF 49.9±13.2% vs. 51.6±8.4%, respectively). The difference lies in the effect in the control mice of these groups, where the EF was markedly lower in the ADAM10 inhibitor series than in the ADAM10 KO series (EF 22.9±9.1% vs. 39.3±5.2%, respectively). This effect has most likely to be attributed to experimental variability, as the EF values achieved for DMSO control mice in the newly added ADAM10 inhibitor series (application of ADAM10 inhibitor for three days after MI in comparison the treatment for 14 d) are comparable with the results of the ADAM10 KO series (EF 32.3±5.7%, **Fig. 4e of the revised manuscript, see below**).

Fig. 4. ADAM10 inhibition reduces neutrophil bone marrow egress and heart tissue infiltration following myocardial infarction. **a** Gating strategy for identification of neutrophils (CD45⁺, CD11b⁺, Ly6G⁺), eosinophils (CD45⁺, CD11b⁺, Ly6G⁻, SigF⁺), monocytes (CD45⁺, CD11b⁺, Ly6G⁻, SigF⁻, F4/80⁻, Ly6C⁺) and macrophages (CD45⁺, CD11b⁺, Ly6G⁻, SigF⁻, F4/80⁺) in the infarcted area/ infarct border zone of A10i (n = 5) and DMSO (n = 4) treated mice 3 days after infarction **b** Analysis of leukocyte counts in blood samples of GI254023X (A10i, n = 5) and DMSO (n = 5) treated mice 3 days after infarction (mean \pm SEM, ns = not significant, **P<0.01, Mann-Whitney test). **c** Analysis of leukocyte counts in the infarcted area/ infarct border zone of GI254023X (A10i, n = 5) and DMSO (n = 5) treated mice 3 days after infarction (mean \pm SEM, ns = not significant, *P<0.05, Mann-Whitney test). **d** Treatment scheme. Following LAD-ligation (MI), mice were treated with GI254023X (A10i, 100 mg/kg) for 3 days (n = 12), 14 days (n = 7) or DMSO (n = 10) using osmotic minipumps. Final echocardiography and sample collection was performed 14 and 28 days after LAD ligation. **e-f** Echocardiography assessment of fractional area shortening, ejection fraction and left ventricular end-systolic interior diameter **e** 14 and **f** 28 days (DMSO, n = 6, A10i - 3 d, n = 11, A10i - 14 d, n = 7) after myocardial infarction (mean \pm SEM, *P<0.05, **P<0.01, one-way ANOVA with Tukey's posttest).

This strongly argues that pharmacologic inhibition has side effects and/or may target other cell types within the heart aside of cardiomyocytes (see above). As to the first possibility there was a clinical trial with GI254023X which was discontinued in phase I/II due to hepatotoxicity following systemic administration (doi: 10.3389/fimmu.2020.00499).

We agree with the Reviewer that GI254023X may have side effects, as has any other drug. Concerning the hepatotoxic side effects of GI254023X the situation is not clear from the literature. The paper

mentioned by the Reviewer (doi: 10.3389/fimmu.2020.00499) cites another review regarding these effects (doi: 10.1152/ajplung.00294.2014) which states: “GI254023X and most other tested compounds are peptidomimetic inhibitors with a reverse hydroxamate group, which chelates the zinc atom in the active site of the protease. Many hydroxamate-based compounds display hepatotoxicity when administered systemically, and clinical trials had to be halted because of these problems.” This seems to refer to a potential class effect of hydroxamate inhibitors and not a specific effect observed for GI254023X. Unfortunately, no further literature is cited. A hepatotoxic effect is still very much possible and will have to be carefully evaluated before GI254023X can be used in the clinical setting. Maybe a non-hydroxamate ADAM10 inhibitor needs to be used. In any case, the very short treatment duration of only three days that we show to be as effective as 14-day treatment with the ADAM10 inhibitor (**Fig. 4d-f of the revised manuscript, see above**) will most likely strongly limit many potential side effects of ADAM10 inhibition. We now elaborate this aspect in the Discussion section of the revised manuscript:

Our data indicate that treatment with A10i for only 3 days post MI may be sufficient for protection from excessive myocardial inflammation, leading to reduced infarct size as well as an expected favorable side effect profile with reduced risk of e.g. hepatotoxicity³⁹ and without prolonged immunosuppression and the risk of severe infection.

c. The authors should consider to reconstruct the manuscript putting the data with the cardio-specific ADAM10 KO in a central position, showing the human data at the end to point out translational potential.

We thank the Reviewer for that suggestion. Indeed, we have now added a high amount of additional data on myocardial infarction in the ADAM10 KO mice to the manuscript, according to the suggestions of the Reviewers. However, we did not repeat every single experiment also in the ADAM10 KO. Therefore, for the reader the succession of experiments in the manuscript still develops in a much more natural and logical way when starting with the ADAM10 inhibitor as opposed to ADAM10 KO. We hope the Reviewer can accept this decision.

Minor

d. It is somewhat disturbing that summary and introduction start with the identical sentence.

We thank the Reviewer for pointing this out. We have changed the first sentence of the Abstract accordingly which now reads as follows:

Myocardial infarction (MI) contributes importantly to cardiac mortality and morbidity.

e. The immunofluorescence images shown in figure 2h require co-staining with cellular markers for cardiomyocytes and non-cardiomyocytes to again prove the localization of ADAM10 in cardiomyocytes.

We agree with the Reviewer and have removed Figure 2h and the corresponding Results section from the revised manuscript.

To further clarify the localization of ADAM10 in heart tissue we have performed immunostaining in cardiac slices of control and MI hearts 14 days after MI. **See response to point a above.**

f. Since the ejection fraction after 28 days in the treated infarct group is not distinguishable from controls while there is still a scar is formed (figure 2 g) this argues that the remaining myocardium most likely is hypertrophied to compensate for the lost cardiomyocytes.

Following the suggestions of the Reviewer we have examined the myocyte cross-sectional area by immunofluorescence in cTnI stained in control and ADAM10 inhibitor treated animals 28 d post MI as well as in sham operated animals. The results show that the ADAM10 inhibitor significantly reduces myocardial hypertrophy. This indicates that the improved cardiac function is not due to larger hypertrophy in the ADAM10 inhibitor treated animals but caused by the reduction in scar size (**Supplementary Fig. 2c of the revised manuscript, see below**). We have added a new section to the Results reading as follows:

Cardiomyocyte cross sectional area was significantly reduced by A10i 28 days after MI compared to controls (Supplementary Fig. 2c).

Supplementary Fig. 2. Pharmacological ADAM10 inhibition augments cardiac function after experimental infarction. a-b Representative end-systolic B-mode and M-mode echocardiograms of GI254023X (A10i) and DMSO treated mice 3 and 14 d after myocardial infarction. **c** Quantification of TNI stained heart tissue sections of GI254023X (A10i) and DMSO treated mice 28 d after myocardial infarction ($n = 4$, mean \pm SEM, $*P < 0.05$, Mann-Whitney test).

g. Data shown in figure 3 are derived from transcriptome analysis of the entire infarcted area. As was discussed above a single cell transcriptomics analysis should have been performed.

The Reviewer has a valid point. The main conclusion that can be drawn from this experiment is indeed that there is less inflammation, but it remains unclear in this experiment which specific cell types are affected. We now have discussed this shortcoming in a new section of the Discussion which reads as follows:

Moreover, single cell transcriptomics would allow a more comprehensive description of the actions of specific cell types in the processes occurring in the infarct and border zone.

h. Figure 6 b , c are basal controls and should go into the appendix

We agree with the Reviewer and have moved Fig. 6b and c to the Supplement as Supplemental Figure 6 b and c.

REVIEWERS' COMMENTS

Reviewer #1 (Remarks to the Author):

The authors have done an excellent job responding to the critique including a de novo study that supports efficacy of the ADAM10 inhibitor one month after only a 3-day exposure post IM.

Reviewer #2 (Remarks to the Author):

The authors have significantly improved their manuscript by adding the requested additional data with ADAM10 cardiomyocyte ko mice and addressing the other comments in a positive and constructive manner.

Responses to Reviewers

Reviewer #1 (Remarks to the Author):

The authors have done an excellent job responding to the critique including a de novo study that supports efficacy of the ADAM10 inhibitor one month after only a 3-day exposure post IM.

We would like to thank the reviewer for his kind comment and are pleased that we were able to address all points to his satisfaction.

Reviewer #2 (Remarks to the Author):

The authors have significantly improved their manuscript by adding the requested additional data with ADAM10 cardiomyocyte ko mice and addressing the other comments in a positive and constructive manner.

We thank the reviewer for his positive comment and are grateful that the results of the requested experiments were to his satisfaction.